# Augmentation-Aware Self-Supervision for Data-Efficient GAN Training

**Liang Hou**[1,3,4], **Qi Cao**[1], **Yige Yuan**[1,3], **Songtao Zhao**[4], **Chongyang Ma**[4],
**Siyuan Pan**[4], **Pengfei Wan**[4], **Zhongyuan Wang**[4], **Huawei Shen**[1,3], **Xueqi Cheng**[2,3]
[1]CAS Key Laboratory of AI Safety and Security,
Institute of Computing Technology, Chinese Academy of Sciences
[2]CAS Key Laboratory of Network Data Science and Technology,
Institute of Computing Technology, Chinese Academy of Sciences
[3]University of Chinese Academy of Sciences   [4]Kuaishou Technology
`lianghou96@gmail.com`

## Abstract

Training generative adversarial networks (GANs) with limited data is challenging because the discriminator is prone to overfitting. Previously proposed differentiable augmentation demonstrates improved data efficiency of training GANs. However, the augmentation implicitly introduces undesired invariance to augmentation for the discriminator since it ignores the change of semantics in the label space caused by data transformation, which may limit the representation learning ability of the discriminator and ultimately affect the generative modeling performance of the generator. To mitigate the negative impact of invariance while inheriting the benefits of data augmentation, we propose a novel augmentation-aware self-supervised discriminator that predicts the augmentation parameter of the augmented data. Particularly, the prediction targets of real data and generated data are required to be distinguished since they are different during training. We further encourage the generator to adversarially learn from the self-supervised discriminator by generating augmentation-predictable real and not fake data. This formulation connects the learning objective of the generator and the arithmetic − harmonic mean divergence under certain assumptions. We compare our method with state-of-the-art (SOTA) methods using the class-conditional BigGAN and unconditional StyleGAN2 architectures on data-limited CIFAR-10, CIFAR-100, FFHQ, LSUN-Cat, and five low-shot datasets. Experimental results demonstrate significant improvements of our method over SOTA methods in training data-efficient GANs.[1]

## 1   Introduction

Generative adversarial networks (GANs) [10] have achieved great progress in synthesizing diverse and high-quality images in recent years [2, 17, 19, 20, 13]. However, the generation quality of GANs depends heavily on the amount of training data [47, 18]. In general, the decrease of training samples usually yields a sharp decline in both fidelity and diversity of the generated images [39, 48]. This issue hinders the wide application of GANs due to the fact of insufficient data in real-world applications. For instance, it is valuable to imitate the style of an artist whose paintings are limited. GANs typically consist of a generator that is designed to generate new data and a discriminator that guides the generator to recover the real data distribution. The major challenge of training GANs under limited data is that the discriminator is prone to overfitting [47, 18], and therefore lacks generalization to teach the generator to learn the underlying real data distribution.

---

[1]Our code is available at `https://github.com/liang-hou/augself-gan`.

37th Conference on Neural Information Processing Systems (NeurIPS 2023).

In order to alleviate the overfitting issue, recent researches have suggested a variety of approaches, mainly from the perspectives of training data [18], loss functions [40], and network architectures [28]. Among them, data augmentation-based methods have gained widespread attention due to its simplicity and extensibility. Specifically, DiffAugment [47] introduced differentiable augmentation techniques for GANs, in which both real and generated data are augmented to supplement the training set of the discriminator. However, this straightforward augmentation method overlooks augmentation-related semantic information, as it solely augments the domain of the discriminator while neglecting the range. Such a practice might introduces an inductive bias that potentially forces the discriminator to remain invariant to different augmentations [24], which could limit the representation learning of the discriminator and subsequently affect the generation performance of the generator [12].

In this paper, we propose a novel augmentation-aware self-supervised discriminator that predicts the augmentation parameter of augmented data with the original data as reference to address the above problem. Meanwhile, the self-supervised discriminator is required to be distinguished between the real data and the generated data since their distributions are different during training, especially in the early stage. The proposed discriminator can benefit the generator in two ways, implicitly and explicitly. On one hand, the self-supervised discriminator can transfer the learned augmentation-aware knowledge to the original discriminator through parameter sharing. On the other hand, we allow the generator to learn adversarially from the self-supervised discriminator by generating augmentation-predictable real and not fake data (Equation (6)). We also theoretically analyzed the connection between this objective function and the minimization of a robust $f$-divergence divergence (the arithmetic $-$ harmonic mean divergence [37]). In experiments, we show that the proposed method compares favorably to the data augmentation counterparts and other state-of-the-art (SOTA) methods on common data-limited benchmarks (CIFAR-10 [21], CIFAR-100 [21], FFHQ [17], LSUN-Cat [45], and five low-shot image generation datasets [36]) based on the class-conditional BigGAN [2] and unconditional StyleGAN2 [19] architectures. In addition, we carried out extensive experiments to demonstrate the effectiveness of the objective function design, the adaptability to stronger data augmentations, and the robustness of hyper-parameter selection in our method.

## 2 Related Work

In this section, we provide an overview of existing work related to training GANs in data-limited scenarios. We also discuss methodologies incorporating self-supervised learning techniques.

### 2.1 GANs under Limited Training Data

Recently, researchers have become interested in freeing training GANs from the need to collect large amounts of data for adaptability in real-world scenarios. Previous studies typically fall into two main categories. The first one involves adopting a pre-trained GAN model to the target domain by fine-tuning partial parameters [41, 31, 42, 30]. However, it requires external training data, and the adoption performance depends heavily on the correlation between the source and target domains.

The other one focuses on training GANs from scratch with elaborated data-efficient training strategies. DiffAugment [47] utilized differentiable augmentation to supplement the training set to prevent discriminator from overfitting in limited data regimes. Concurrently, ADA [18] introduced adaptive data augmentation with a richer set of augmentation categories. APA [16] adaptively augmented the real data with the most plausible generated data. LeCam-GAN [40] proposed adaptive regularization for the discriminator and showed a connection to the Le Cam divergence [23]. [3] discovered that sparse sub-network (lottery tickets) [5] and feature-level adversarial augmentation could offer orthogonal gains to data augmentation methods. InsGen [43] improved the data efficiency of training GANs by incorporating instance discrimination tasks to the discriminator. MaskedGAN [14] employed masking in the spatial and spectral domains to alleviate the discriminator overfitting issue. GenCo [7] discriminated samples from multiple views with weight-discrepancy and data-discrepancy mechanisms. FreGAN [44] focused on discriminating between real and fake samples in the high-frequency domain. DigGAN [8] constrains the discriminator gradient gap between real and generated data. FastGAN [28] designed a lightweight generator architecture and observed that a self-supervised discriminator could enhance low-shot generation performance. Our method falls into the second category, supplementing data augmentation-based GANs and can be also applied to other methods.

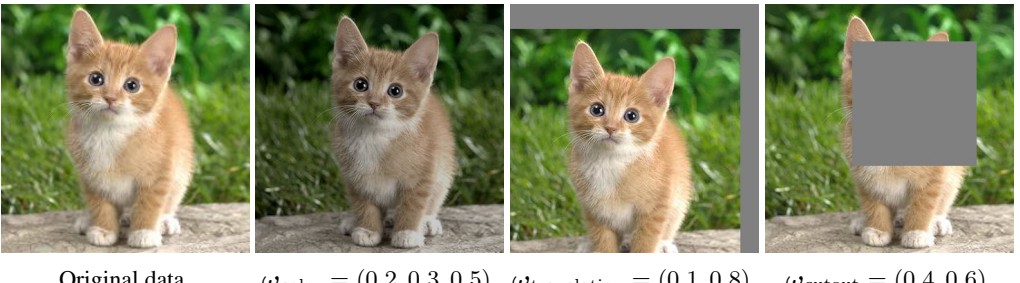

| Original data | $\boldsymbol{\omega}_{\text{color}} = (0.2, 0.3, 0.5)$ | $\boldsymbol{\omega}_{\text{translation}} = (0.1, 0.8)$ | $\boldsymbol{\omega}_{\text{cutout}} = (0.4, 0.6)$ |

Figure 1: Examples of images with different kinds of differentiable augmentation (including the original unaugmented one) and their re-scaled corresponding augmentation parameters $\boldsymbol{\omega} \in [0, 1]^d$.

## 2.2 GANs with Self-Supervised Learning

Self-supervised learning techniques excel at learning meaningful representations without human-annotated labels by solving pretext tasks. Transformation-based self-supervised learning methods such as rotation recognition [9] have been incorporated into GANs to address catastrophic forgetting in discriminators [4, 38, 12]. Various other self-supervised tasks have also been explored, including jigsaw puzzle solving [1], latent transformation detection [33], and mutual information maximization [25]. Moreover, ContraD [15] decouples the representation learning and discrimination of the discriminator, utilizing contrastive learning for representation learning and a discriminator head for distinguishing real from fake upon the contrastive representations. In contrast to ours, CR-GAN [46] and ICR-GAN proposed consistency regularization for the discriminator, which corresponds to an explicit augmentation-invariant of the discriminator. both our proposed method and SSGAN-LA [12] belong to adversarial self-supervised learning, they differ in the type of self-supervised signals and model inputs. SSGAN-LA is limited to categorical self-supervision [9], which is incompatible with popular augmentation-based GANs like DiffAugment [47]. Our method is applicable for continuous self-supervision and integrates seamlessly with DiffAugment. Furthermore, continuous self-supervision have a magnitude relationship and thus can provide more refined gradient feedback for the model to overcome overfitting in data-limited scenarios. Additionally, unlike SSGAN-LA, our method does not constrain the invertibility of data transformations (Theorem 1) because it additionally take the original sample as input for the self-supervised discriminator (Equation (5)).

## 3 Preliminaries

In this section, we introduce the necessary concepts and preliminaries for completeness of the paper.

### 3.1 Generative Adversarial Networks

Generative adversarial networks (GANs) [10] typically contain a generator $G : \mathcal{Z} \to \mathcal{X}$ that maps a low-dimensional latent code $\mathbf{z} \in \mathcal{Z}$ endowed with a tractable prior $p(\mathbf{z})$, e.g., multivariate normal distribution $\mathcal{N}(\mathbf{0}, \mathbf{I})$, to a high-dimensional data point $\mathbf{x} \in \mathcal{X}$, which induces a generated data distribution (density) $p_G(\mathbf{x}) = \int_{\mathcal{Z}} p(\mathbf{z}) \delta(\mathbf{x} - G(\mathbf{z})) d\mathbf{z}$ with the Dirac delta distribution $\delta(\cdot)$, and also contain a discriminator $D : \mathcal{X} \to \mathbb{R}$ that is required to distinguish between the real data sampled from the underlying data distribution (density) $p_{\text{data}}(\mathbf{x})$ and the generated ones. The generator attempts to fool the discriminator to eventually recover the real data distribution, i.e., $p_G(\mathbf{x}) = p_{\text{data}}(\mathbf{x})$. Formally, the loss functions for the discriminator and the generator can be formulated as follows:

$$\mathcal{L}_D = \mathbb{E}_{\mathbf{x} \sim p_{\text{data}}(\mathbf{x})}[f(D(\mathbf{x}))] + \mathbb{E}_{\mathbf{z} \sim p(\mathbf{z})}[h(D(G(\mathbf{z})))], \tag{1}$$

$$\mathcal{L}_G = \mathbb{E}_{\mathbf{z} \sim p(\mathbf{z})}[g(D(G(\mathbf{z})))]. \tag{2}$$

Different real-valued functions $f$, $h$, and $g$ correspond to different variants of GANs [32]. For example, the minimax GAN [10] can be constructed by setting $f(x) = -\log(\sigma(x))$ and $h(x) = -g(x) = -\log(1 - \sigma(x))$ with the sigmoid function $\sigma(x) = 1/(1 + \exp(-x))$. In this study, we follow the practices of DiffAugment [47] to adopt the hinge loss [26], i.e., $f(x) = h(-x) = \max(0, 1 - x)$ and $g(x) = -x$, for experiments based on BigGAN [2] and the log loss [10], i.e., $f(x) = g(x) = -\log(\sigma(x))$ and $h(x) = -\log(1 - \sigma(x))$, for experiments based on StyleGAN2 [19].

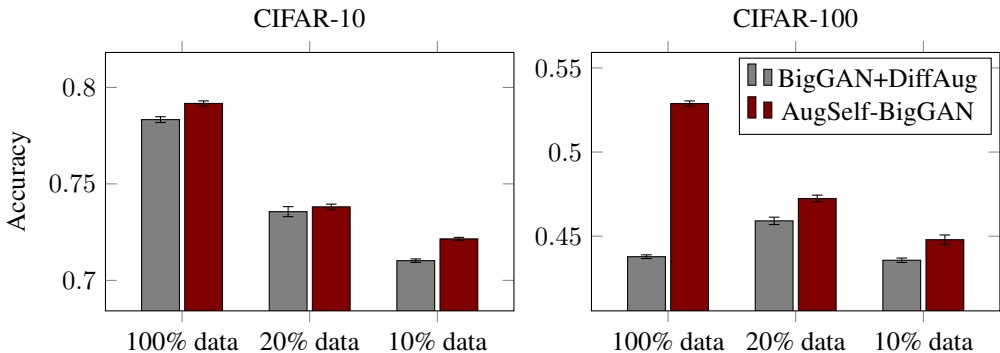

Figure 2: Comparison of representation learning ability of discriminator between BigGAN + DiffAugment and our AugSelf-BigGAN on CIFAR-10 and CIFAR-100 using linear logistic regression.

## 3.2 Differentiable Augmentation for GANs

DiffAugment [47] introduces differentiable augmentation $T : \mathcal{X} \times \Omega \rightarrow \hat{\mathcal{X}}$ parameterized by a randomly-sampled parameter $\boldsymbol{\omega} \in \Omega$ with a prior $p(\boldsymbol{\omega})$ for data-efficient GAN training. The parameter $\boldsymbol{\omega}$ determines exactly how to transfer a sample $\mathbf{x}$ to an augmented one $\hat{\mathbf{x}} \in \hat{\mathcal{X}}$ for the discriminator. After manually re-scaling (for $\boldsymbol{\omega} \in [0,1]^d$), the parameters of all three kinds of differentiable augmentation used in DiffAugment for 2D images can be expressed as follows:

- color: $\boldsymbol{\omega}_{\text{color}} = (\lambda_{\text{brightness}}, \lambda_{\text{saturation}}, \lambda_{\text{contrast}}) \in [0,1]^3$;
- translation: $\boldsymbol{\omega}_{\text{translation}} = (x_{\text{translation}}, y_{\text{translation}}) \in [0,1]^2$;
- cutout: $\boldsymbol{\omega}_{\text{cutout}} = (x_{\text{offset}}, y_{\text{offset}}) \in [0,1]^2$.

Figure 1 illustrates the augmentation operations and their parameters. Formally, the loss functions for the discriminator and the generator of GANs with DiffAugment are defined as follows:

$$\mathcal{L}_D^{\text{da}} = \mathbb{E}_{\mathbf{x} \sim p_{\text{data}}(\mathbf{x}), \boldsymbol{\omega} \sim p(\boldsymbol{\omega})}[f(D(T(\mathbf{x}; \boldsymbol{\omega})))] + \mathbb{E}_{\mathbf{z} \sim p(\mathbf{z}), \boldsymbol{\omega} \sim p(\boldsymbol{\omega})}[h(D(T(G(\mathbf{z}); \boldsymbol{\omega})))], \quad (3)$$

$$\mathcal{L}_G^{\text{da}} = \mathbb{E}_{\mathbf{z} \sim p(\mathbf{z}), \boldsymbol{\omega} \sim p(\boldsymbol{\omega})}[g(D(T(G(\mathbf{z}); \boldsymbol{\omega})))], \quad (4)$$

where $\boldsymbol{\omega}$ can represent any combination of these parameters. We choose all augmentations by default, which means augmentation color, translation, and cutout are adopted for each image sequentially.

## 4 Method

Data augmentation for GANs allows the discriminator to distinguish a single sample from multiple perspectives by transforming it into various augmented samples according to different augmentation parameters. However, it overlooks the differences in augmentation intensity, such as color contrast and translation magnitude, leading the discriminator to implicitly maintain invariance to these varying intensities. The invariance may limit the representation learning ability of the discriminator because it loses augmentation-related information (e.g., color and position) [24]. Figure 2 confirms the impact of this point on the discriminator representation learning task [4]. We argue that a discriminator that captures comprehensive representations contributes to better convergence of the generator [35, 22]. Moreover, data augmentation may lead to augmentation leaking in generated data, when using specific data augmentations such as random 90-degree rotations [18, 12]. Therefore, our goal is to eliminate the unnecessary potential inductive bias (invariance to augmentations) for the discriminator while preserving the benefits of data augmentation for training data-efficient GANs.

To achieve this goal, we propose a novel augmentation-aware self-supervised discriminator $\hat{D} : \hat{\mathcal{X}} \times \mathcal{X} \rightarrow \Omega^+ \cup \Omega^-$ that predicts the augmentation parameter and authenticity of the augmented data given the original data as reference. Distinguishing between the real data and the generated data with different self-supervision is because they are different during training, especially in the early stage. Specifically, the predictive targets of real data and generated data are represented as $\boldsymbol{\omega}^+ \in \Omega^+$ and $\boldsymbol{\omega}^- \in \Omega^-$, respectively. They are constructed from the augmentation parameter $\boldsymbol{\omega}$ with different transformations, i.e., $\boldsymbol{\omega}^+ = -\boldsymbol{\omega}^- = \boldsymbol{\omega}$. Since the augmentation parameter is a continuous

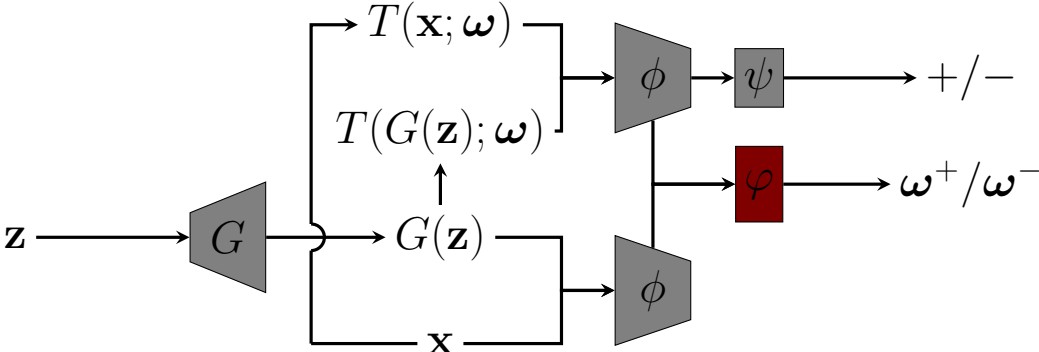

Figure 3: Diagram of AugSelf-GAN. The original augmentation-based discriminator is $D(T(\cdot)) = \psi(\phi(T(\cdot)))$. The augmentation-aware self-supervised discriminator is $\hat{D}(T(\cdot), \cdot) = \varphi(\phi(T(\cdot)) - \phi(\cdot))$, where $\varphi$ is our newly introduced linear layer with negligible additional parameters.

vector, we use mean squared error loss to regress it. The proposed method combines continuous self-supervised signals with real-vs-fake discrimination signals, thus can be considered as soft-label augmentation [12]. Comparison with self-supervision that does not distinguish between real and fake is referred to Table 6 in Appendix C. Notice that the predictive targets (augmentations) can be a subset of performed augmentations (see Table 7 in Appendix C for comparison). Mathematically, the loss function for the augmentation-aware self-supervised discriminator is formulated as the following:

$$\mathcal{L}_{\hat{D}}^{\text{ss}} = \mathbb{E}_{\mathbf{x},\boldsymbol{\omega}} \left[ \|\hat{D}(T(\mathbf{x};\boldsymbol{\omega}), \mathbf{x}) - \boldsymbol{\omega}^+\|_2^2 \right] + \mathbb{E}_{\mathbf{z},\boldsymbol{\omega}} \left[ \|\hat{D}(T(G(\mathbf{z});\boldsymbol{\omega}), G(\mathbf{z})) - \boldsymbol{\omega}^-\|_2^2 \right]. \quad (5)$$

In our implementations, the proposed self-supervised discriminator $\hat{D} = \varphi \circ \phi$ shares the backbone $\phi : \mathcal{X} \to \mathbb{R}^d$ with the original discriminator $D = \psi \circ \phi$ except the output linear layer $\varphi : \mathbb{R}^d \to \Omega^+ \cup \Omega^-$. This parameter-sharing design not only improves the representation learning ability of the original discriminator but also saves the number of parameters in our model compared to the base model, e.g., $0.04\%$ more parameters in BigGAN and $0.01\%$ in StyleGAN2. More specifically, the self-supervised discriminator predicts the target based on the difference between learned representations of the augmented data and the original data, i.e., $\hat{D}(T(\mathbf{x};\boldsymbol{\omega}), \mathbf{x}) = \varphi(\phi(T(\mathbf{x};\boldsymbol{\omega})) - \phi(\mathbf{x}))$ (see Table 8 in Appendix C for comparison with other architectures). The philosophy behind our design is that the backbone $\phi$ should capture rich (which necessitates the design of a simple head $\varphi$) and linear (inspiring us to perform subtraction on the features) representations.

In order for the generator to directly benefit from the self-supervision of data augmentation, we establish a novel adversarial game between the augmentation-aware self-supervised discriminator and the generator with the objective function for the generator defined as follows:

$$\mathcal{L}_G^{\text{ss}} = \mathbb{E}_{\mathbf{z},\boldsymbol{\omega}} \left[ \|\hat{D}(T(G(\mathbf{z});\boldsymbol{\omega}), G(\mathbf{z})) - \boldsymbol{\omega}^+\|_2^2 \right] - \mathbb{E}_{\mathbf{z},\boldsymbol{\omega}} \left[ \|\hat{D}(T(G(\mathbf{z});\boldsymbol{\omega}), G(\mathbf{z})) - \boldsymbol{\omega}^-\|_2^2 \right]. \quad (6)$$

The objective function is actually the combination of the non-saturating loss (regarding the generated data as real, $\min_G \mathbb{E}_{\mathbf{z},\boldsymbol{\omega}}[\|\hat{D}(T(G(\mathbf{z});\boldsymbol{\omega}), G(\mathbf{z})) - \boldsymbol{\omega}^+\|_2^2]$) and the saturating loss (reversely optimizing the objective function of the discriminator, $\max_G \mathbb{E}_{\mathbf{z},\boldsymbol{\omega}}[\|\hat{D}(T(G(\mathbf{z});\boldsymbol{\omega}), G(\mathbf{z})) - \boldsymbol{\omega}^-\|_2^2]$) (see Table 9 in Appendix C for ablation). Intuitively, the non-saturating loss encourages the generator to produce augmentation-predictable data, facilitating fidelity but reducing diversity. Conversely, the saturating loss strives for the generator to avoid generating augmentation-predictable data, promoting diversity at the cost of fidelity. We will elucidate in Section 5 how this formalization assists the generator in matching the fidelity and diversity of real data, ultimately leading to an accurate approximation of the target data distribution.

The total objective functions for the original discriminator, the augmentation-aware self-supervised discriminator, and the generator of our proposed method, named AugSelf-GAN, are given by:

$$\min_{D,\hat{D}} \mathcal{L}_D^{\text{da}} + \lambda_d \cdot \mathcal{L}_{\hat{D}}^{\text{ss}}, \quad (7)$$

$$\min_G \mathcal{L}_G^{\text{da}} + \lambda_g \cdot \mathcal{L}_G^{\text{ss}}, \quad (8)$$

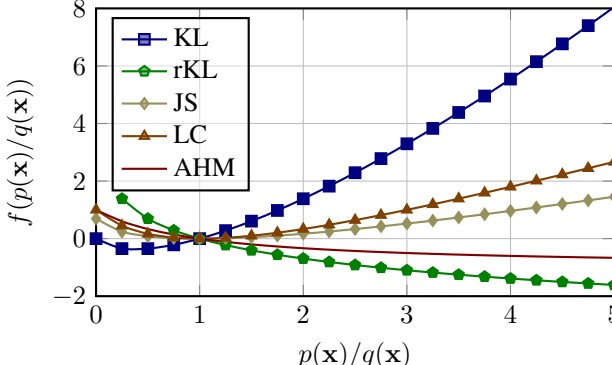

Figure 4: Comparison of the function $f$ in different $f$-divergences. The $f$-divergence between two probability distributions $p(\mathbf{x})$ and $q(\mathbf{x})$ is defined as $D_f(p(\mathbf{x})\|q(\mathbf{x})) = \int_{\mathcal{X}} q(\mathbf{x})f(p(\mathbf{x})/q(\mathbf{x}))\mathrm{d}\mathbf{x}$ with a convex function $f : \mathbb{R}_{\geq 0} \to \mathbb{R}$ satisfying $f(1) = 0$. The x- and y-axis denote the input and the value of the function $f$ in the $f$-divergence. The function $f$ of the AHM divergence yields the most robust value for large inputs.

where the hyper-parameters are set as $\lambda_d = \lambda_g = 1$ in experiments by default unless otherwise specified (see Figure 6 for empirical studies). Details of objective functions are referred to Appendix B.

## 5 Theoretical Analysis

In this section, we analyze the connection between the theoretical learning objective of AugSelf-GAN and the arithmetic − harmonic mean (AHM) divergence [37] under certain assumptions.

**Proposition 1.** *For any generator $G$ and given unlimited capacity in the function space, the optimal augmentation-aware self-supervised discriminator $\hat{D}^*$ has the form of:*

$$\hat{D}^*(\hat{\mathbf{x}}, \mathbf{x}) = \frac{\int p_{\mathrm{data}}(\mathbf{x}, \boldsymbol{\omega}, \hat{\mathbf{x}})\boldsymbol{\omega}^+\mathrm{d}\boldsymbol{\omega} + \int p_G(\mathbf{x}, \boldsymbol{\omega}, \hat{\mathbf{x}})\boldsymbol{\omega}^-\mathrm{d}\boldsymbol{\omega}}{p_{\mathrm{data}}(\mathbf{x}, \hat{\mathbf{x}}) + p_G(\mathbf{x}, \hat{\mathbf{x}})} \tag{9}$$

The proofs of all theoretical results (including the following ones) are deferred in Appendix A.

**Theorem 1.** *Assume that $\boldsymbol{\omega}^+ = -\boldsymbol{\omega}^- = \mathbf{c}$ is constant, under the optimal self-supervised discriminator $\hat{D}^*$, optimizing the self-supervised task for the generator $G$ is equivalent to:*

$$\min_G 4c \cdot M_{\mathrm{AH}}(p_{\mathrm{data}}(\mathbf{x}, \hat{\mathbf{x}})\|p_G(\mathbf{x}, \hat{\mathbf{x}})), \tag{10}$$

*where $c = \|\mathbf{c}\|_2^2$ is constant and $M_{\mathrm{AH}}$ is the arithmetic − harmonic mean divergence [37], of which the minimum is achieved if and only if $p_G(\mathbf{x}, \hat{\mathbf{x}}) = p_{\mathrm{data}}(\mathbf{x}, \hat{\mathbf{x}}) \Rightarrow p_G(\mathbf{x}) = p_{\mathrm{data}}(\mathbf{x})$.*

Theorem 1 reveals that the generator of AugSelf-GAN theoretically still satisfies generative modeling, i.e., accurately learning the target data distribution, under certain assumptions. Although AugSelf-GAN does not obey the strict assumption, we note that this is not rare in the literature.[2] Under this assumption, AugSelf-GAN can be regarded as a multi-dimensional extension of LS-GAN [29] in terms of the loss function, while excluding that of the generator. Additionally, our analysis offers an alternative theoretically grounded generator loss function for the LS-GAN family.[3]

**Corollary 1.** *The following equality and inequality hold for the AHM divergence:*

- $M_{\mathrm{AH}}(p_{\mathrm{data}}(\mathbf{x}, \hat{\mathbf{x}})\|p_G(\mathbf{x}, \hat{\mathbf{x}})) + M_{\mathrm{AH}}(p_G(\mathbf{x}, \hat{\mathbf{x}})\|p_{\mathrm{data}}(\mathbf{x}, \hat{\mathbf{x}})) = \Delta(p_{\mathrm{data}}(\mathbf{x}, \hat{\mathbf{x}})\|p_G(\mathbf{x}, \hat{\mathbf{x}}))$
- $M_{\mathrm{AH}}(p_{\mathrm{data}}(\mathbf{x}, \hat{\mathbf{x}})\|p_G(\mathbf{x}, \hat{\mathbf{x}})) = 1 - W(p_{\mathrm{data}}(\mathbf{x}, \hat{\mathbf{x}})\|p_G(\mathbf{x}, \hat{\mathbf{x}})) \leq 1$

*where $\Delta$ is the Le Cam (LC) divergence [23], and $W$ is the harmonic mean divergence [37].*

Corollary 1 reveals an inequality $M_{\mathrm{AH}}(p_{\mathrm{data}}(\mathbf{x}, \hat{\mathbf{x}})\|p_G(\mathbf{x}, \hat{\mathbf{x}})) \leq \Delta(p_{\mathrm{data}}(\mathbf{x}, \hat{\mathbf{x}})\|p_G(\mathbf{x}, \hat{\mathbf{x}}))$ because of non-negativity of AHM divergence $M_{\mathrm{AH}}(p_G(\mathbf{x}, \hat{\mathbf{x}})\|p_{\mathrm{data}}(\mathbf{x}, \hat{\mathbf{x}})) \geq 0$. Figure 4 plots the function $f$ in AHM divergence and other common $f$-divergences in the GAN literature. The AHM divergence shows better robustness of the function $f$ than others for extremely large inputs $p(\mathbf{x})/q(\mathbf{x}) = D^*(\mathbf{x})/(1 - D^*(\mathbf{x}))$, which is likely for the optimal discriminator $D^*$ in data-limited scenarios.

---

[2]Goodfellow et al. [10] analyzed that saturated GAN optimizes the Jensen Shannon (JS) divergence, but in fact it uses non-saturated loss. LeCam-GAN [40] showed a connection between the Le Cam (LC) divergence [23] and its objective function based on fixed regularization, but in practice it uses exponential moving average.

[3]The theoretical analysis of LS-GAN is actually inconsistent with its generator loss function.

Table 1: IS and FID comparisons of AugSelf-BigGAN with state-of-the-art methods on CIFAR-10 and CIFAR-100 with full and limited data. The best result is **bold** and the second best is underlined.

| | Method | 100% training data | | 20% training data | | 10% training data | |
|---|---|---|---|---|---|---|---|
| | | IS ($\uparrow$) | FID ($\downarrow$) | IS ($\uparrow$) | FID ($\downarrow$) | IS ($\uparrow$) | FID ($\downarrow$) |
| CIFAR-10 | BigGAN [2] | 9.07 | 9.59 | 8.52 | 21.58 | 7.09 | 39.78 |
| | DiffAugment [47] | 9.16 | 8.70 | 8.65 | 14.04 | 8.09 | 22.40 |
| | CR-GAN [46] | 9.17 | 8.49 | 8.61 | 12.84 | 8.49 | 18.70 |
| | LeCam-GAN [40] | **9.43** | 8.28 | 8.83 | 12.56 | 8.57 | 17.68 |
| | DigGAN [8] | 9.28 | 8.49 | 8.89 | 13.01 | 8.32 | 17.87 |
| | Tickets [3] | - | 8.19 | - | 12.83 | - | 16.74 |
| | MaskedGAN [14] | - | $8.41_{\pm.03}$ | - | $12.51_{\pm.09}$ | - | $15.89_{\pm.12}$ |
| | GenCo [7] | - | $7.98_{\pm.02}$ | - | $12.61_{\pm.05}$ | - | $18.10_{\pm.06}$ |
| | AugSelf-BigGAN | $\mathbf{9.43}_{\pm.14}$ | $7.68_{\pm.06}$ | $8.98_{\pm.09}$ | $10.97_{\pm.09}$ | $8.76_{\pm.05}$ | $15.68_{\pm.26}$ |
| | AugSelf-BigGAN+ | $9.27_{\pm.05}$ | $\mathbf{7.54}_{\pm.04}$ | $\mathbf{9.08}_{\pm.04}$ | $\mathbf{9.95}_{\pm.17}$ | $\mathbf{8.79}_{\pm.04}$ | $\mathbf{12.76}_{\pm.14}$ |
| CIFAR-100 | BigGAN [2] | 10.71 | 12.87 | 8.58 | 33.11 | 6.74 | 66.71 |
| | DiffAugment [47] | 10.66 | 12.00 | 9.47 | 22.14 | 8.38 | 33.70 |
| | CR-GAN [46] | 10.81 | 11.25 | 9.12 | 20.28 | 8.70 | 26.90 |
| | LeCam-GAN [40] | 11.05 | 11.20 | 9.81 | 18.03 | 9.27 | 27.63 |
| | DigGAN [8] | **11.45** | 11.63 | 9.54 | 19.79 | 8.98 | 24.59 |
| | Tickets [3] | - | 10.73 | - | 17.43 | - | 23.80 |
| | MaskedGAN [14] | - | $11.65_{\pm.03}$ | - | $18.33_{\pm.09}$ | - | $24.02_{\pm.12}$ |
| | GenCo [7] | - | $10.92_{\pm.02}$ | - | $18.44_{\pm.04}$ | - | $25.22_{\pm.06}$ |
| | AugSelf-BigGAN | $11.19_{\pm.09}$ | $\mathbf{9.88}_{\pm.07}$ | $\mathbf{10.25}_{\pm.06}$ | $16.11_{\pm.25}$ | $9.78_{\pm.08}$ | $21.30_{\pm.15}$ |
| | AugSelf-BigGAN+ | $11.12_{\pm.10}$ | $10.09_{\pm.05}$ | $10.14_{\pm.11}$ | $\mathbf{15.33}_{\pm.20}$ | $\mathbf{9.93}_{\pm.06}$ | $\mathbf{18.64}_{\pm.09}$ |

Table 2: FID comparison of AugSelf-StyleGAN2 with competing methods on FFHQ and LSUN-Cat with limited training samples. The best result is highlighted in **bold** and the second best is underlined.

| Method | FFHQ | | | | LSUN-Cat | | | |
|---|---|---|---|---|---|---|---|---|
| | 30K | 10K | 5K | 1K | 30K | 10K | 5K | 1K |
| StyleGAN2 [19] | 6.16 | 14.75 | 26.60 | 62.16 | 10.12 | 17.93 | 34.69 | 182.85 |
| + ADA [18] | 5.46 | 8.13 | 10.96 | 21.29 | 10.50 | 13.13 | 16.95 | 43.25 |
| + DiffAugment [47] | 5.05 | 7.86 | 10.45 | 25.66 | 9.68 | 12.07 | 16.11 | 42.26 |
| AugSelf-StyleGAN2 | **4.95** | 6.98 | 9.69 | 23.38 | **9.22** | **11.98** | 14.86 | 36.76 |
| AugSelf-StyleGAN2+ | 5.82 | **6.65** | **9.15** | **20.39** | 9.43 | 12.00 | **14.12** | **26.52** |

## 6 Experiments

We implement AugSelf-GAN based on DiffAugment [47], keeping the backbones and settings unchanged for fair comparisons with prior work under two evaluation metrics, IS [34] and FID [11]. The mean and standard deviation (if reported) are obtained with five evaluation runs at the best FID checkpoint. Each of experiments in this work was conducted on an 32GB NVIDIA V100 GPU.

### 6.1 Comparison with State-of-the-Art Methods

**CIFAR-10 and CIFAR-100.** Table 1 reports the results on CIFAR-10 and CIFAR-100 [21]. These experiments are based on the BigGAN architecture [2]. Our method significantly outperforms the direct baseline DiffAugment [47] and yields the best generation performance in terms of FID and IS compared with SOTA methods. Notably, our method achieves further improvement when using stronger augmentation (see Table 5), i.e., AugSelf-BigGAN+ (translation$\uparrow$ and cutout$\uparrow$).

**FFHQ and LSUN-Cat.** Table 2 reports the FID results on FFHQ [17] and LSUN-Cat [45]. The hyper-parameter is $\lambda_g = 0.2$. AugSelf-GAN performs substantially better than baselines with the same network backbone. Also, stronger augmentation, i.e., AugSelf-StyleGAN2+ (translation$\uparrow$ and cutout$\uparrow$), further improves the performance when training data is very limited.

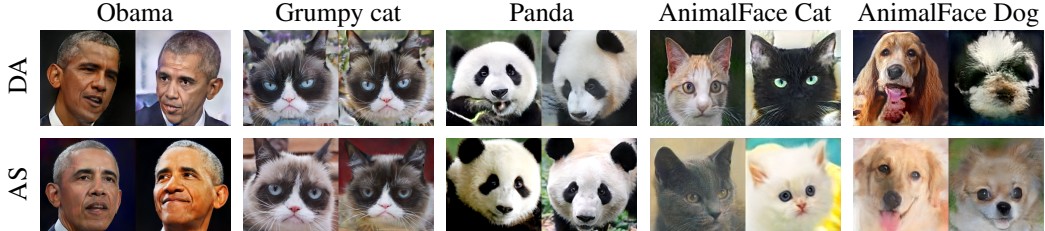

Figure 5: Qualitative comparison between DiffAug (DA) and AugSelf (AS) on low-shot generation.

Table 3: FID comparison of AugSelf-StyleGAN2 with state-of-the-art methods with and without pre-training on five low-shot datasets. The best result is in **bold** and the second best is underlined.

| Method | Pre-training? | 100-shot | | | AnimalFaces | |
| | | Obama | Grumpy cat | Panda | Cat | Dog |
|---|---|---|---|---|---|---|
| Scale/shift [31] | Yes | 50.72 | 34.20 | 21.38 | 54.83 | 83.04 |
| MineGAN [42] | Yes | 50.63 | 34.54 | 14.84 | 54.45 | 93.03 |
| TransferGAN [41] | Yes | 48.73 | 34.06 | 23.20 | 52.61 | 82.38 |
| FreezeD [30] | Yes | 41.87 | 31.22 | 17.95 | 47.70 | 70.46 |
| StyleGAN2 [19] | No | 80.20 | 48.90 | 34.27 | 71.71 | 130.19 |
| + AdvAug [3] | No | 52.86 | 31.02 | 14.75 | 47.40 | 68.28 |
| + ADA [18] | No | 45.69 | 26.62 | 12.90 | 40.77 | 56.83 |
| + APA [16] | No | 42.97 | 28.10 | 19.21 | 42.60 | 81.16 |
| + DiffAugment [47] | No | 46.87 | 27.08 | 12.06 | 42.44 | 58.85 |
| AugSelf-StyleGAN2 | No | **26.00** | **19.81** | **8.36** | **30.53** | **48.19** |

**Low-shot image generation.** Table 3 shows the FID scores on five common low-shot image generation benchmarks [36] (Obama, Grumpy cat, Panda, AnimalFace cat, and AnimalFace dog). The baselines are divided into two categories according to whether they were pre-trained. Due to its training stability, we can train AugSelf-StyleGAN2 for 5k generator update steps to ensure convergence. The hyper-parameters are $\lambda_d = \lambda_g = 0.1$ on Grumpy cat and AnimalFace cat, and the self-supervision is color on all datasets. Impressively, AugSelf-StyleGAN2 surpasses competing methods by a large margin on all low-shot datasets, approaching SOTA FID scores to our knowledge. Figure 5 visually compares the generated images of DiffAugment and AugSelf-GAN, revealing the latter as superior in generating images with more diversity and fewer artifacts.

## 6.2 Analysis of AugSelf-GAN

**Fixed supervision.** Table 4 reports the results of AugSelf-GAN on CIFAR-10 and CIFAR-100 compared to the setup using fixed self-supervision, i.e., $\omega^+ = -\omega^- = 1$, which corresponds to the assumption in Theorem 1. AugSelf-GAN outperforms the fixed self-supervision in terms of both IS and FID in all training data regimes. The reason is somewhat intuitive, as fixed self-supervision does not constitute self-supervised learning for the model, therefore it cannot enable the discriminator to learn semantic information related to data augmentation to optimize the generator. Interestingly, the fixed one beats the baseline, which may be attributed to the multi-input [27] and regression loss [29].

Table 4: FID comparison with fixed self-supervision. The best results are highlighted in **bold**.

| | Method | 100% training data | | 20% training data | | 10% training data | |
| | | IS ($\uparrow$) | FID ($\downarrow$) | IS ($\uparrow$) | FID ($\downarrow$) | IS ($\uparrow$) | FID ($\downarrow$) |
|---|---|---|---|---|---|---|---|
| C10 | Fixed ($c=1$) | $9.25_{\pm.17}$ | $8.01_{\pm.05}$ | $8.70_{\pm.12}$ | $12.58_{\pm.16}$ | $8.53_{\pm.05}$ | $17.66_{\pm.55}$ |
| | AugSelf-GAN | $\mathbf{9.43}_{\pm.14}$ | $\mathbf{7.68}_{\pm.06}$ | $\mathbf{8.98}_{\pm.09}$ | $\mathbf{10.97}_{\pm.09}$ | $\mathbf{8.76}_{\pm.05}$ | $\mathbf{15.68}_{\pm.26}$ |
| C100 | Fixed ($c=1$) | $10.67_{\pm.06}$ | $12.02_{\pm.07}$ | $9.94_{\pm.06}$ | $17.70_{\pm.17}$ | $9.50_{\pm.13}$ | $22.84_{\pm.28}$ |
| | AugSelf-GAN | $\mathbf{11.19}_{\pm.09}$ | $\mathbf{9.88}_{\pm.07}$ | $\mathbf{10.25}_{\pm.06}$ | $\mathbf{16.11}_{\pm.25}$ | $\mathbf{9.78}_{\pm.08}$ | $\mathbf{21.30}_{\pm.15}$ |

Table 5: Study on stronger augmentation. The best is in **bold** and the second best is underlined.

| | Method | 100% training data | | 20% training data | | 10% training data | |
|---|---|---|---|---|---|---|---|
| | | IS ($\uparrow$) | FID ($\downarrow$) | IS ($\uparrow$) | FID ($\downarrow$) | IS ($\uparrow$) | FID ($\downarrow$) |
| C10 | DiffAugment | 9.29$_{\pm.02}$ | 8.48$_{\pm.13}$ | 8.84$_{\pm.12}$ | 15.14$_{\pm.47}$ | **8.80**$_{\pm.01}$ | 20.60$_{\pm.13}$ |
| | + trans.$\uparrow$ cut.$\uparrow$ | 9.28$_{\pm.06}$ | 8.42$_{\pm.18}$ | 8.78$_{\pm.06}$ | 14.28$_{\pm.27}$ | 8.69$_{\pm.07}$ | 20.93$_{\pm.21}$ |
| | AugSelf-GAN | **9.43**$_{\pm.14}$ | 7.68$_{\pm.06}$ | 8.98$_{\pm.09}$ | 10.97$_{\pm.09}$ | 8.76$_{\pm.05}$ | 15.68$_{\pm.26}$ |
| | + trans.$\uparrow$ cut.$\uparrow$ | 9.27$_{\pm.05}$ | **7.54**$_{\pm.04}$ | **9.08**$_{\pm.04}$ | **9.95**$_{\pm.17}$ | 8.79$_{\pm.04}$ | **12.76**$_{\pm.14}$ |
| C100 | DiffAugment | 11.02$_{\pm.07}$ | 11.49$_{\pm.21}$ | 9.45$_{\pm.05}$ | 24.98$_{\pm.48}$ | 8.50$_{\pm.09}$ | 34.92$_{\pm.63}$ |
| | + trans.$\uparrow$ cut.$\uparrow$ | 11.10$_{\pm.08}$ | 11.28$_{\pm.20}$ | 9.58$_{\pm.05}$ | 24.10$_{\pm.66}$ | 8.59$_{\pm.04}$ | 35.32$_{\pm.46}$ |
| | AugSelf-GAN | **11.19**$_{\pm.09}$ | **9.88**$_{\pm.07}$ | **10.25**$_{\pm.06}$ | 16.11$_{\pm.25}$ | 9.78$_{\pm.08}$ | 21.30$_{\pm.15}$ |
| | + trans.$\uparrow$ cut.$\uparrow$ | 11.12$_{\pm.10}$ | 10.09$_{\pm.05}$ | 10.14$_{\pm.11}$ | **15.33**$_{\pm.20}$ | **9.93**$_{\pm.06}$ | **18.64**$_{\pm.09}$ |

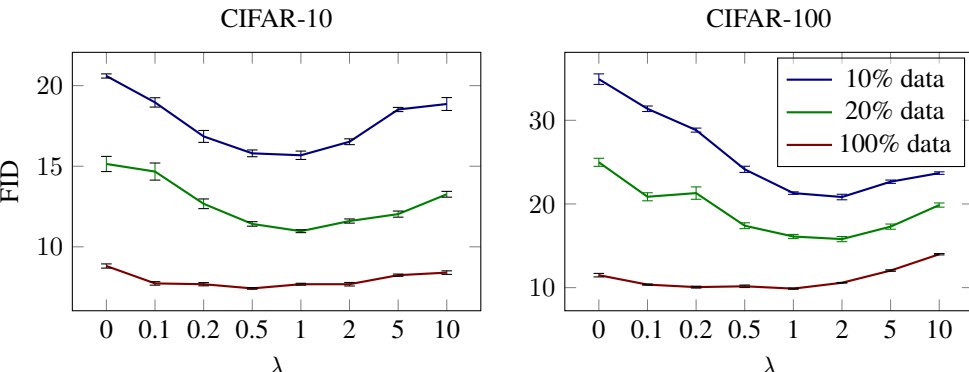

Figure 6: FID curves with varying hyper-parameters $\lambda = \lambda_d = \lambda_g \in [0, 10]$ on CIFAR-10 and CIFAR-100. The hyper-parameter $\lambda = 0$ corresponds to the baseline BigGAN + DiffAugment.

**Stronger augmentation.** Translation and cutout actually erase parts of image information, which help prevent the discriminator from overfitting, but could suffer from underfitting if excessive. Our self-supervised task enables the discriminator to be aware of different levels of translation and cutout, which helps alleviate underfitting and allows us to explore stronger translation and cutout. Table 5 compares AugSelf-GAN with DiffAugment in this setting. Overall, when data is limited, AugSelf-GAN can further benefit from stronger translation and cutout and achieve new SOTA FID results, while DiffAugment cannot. This implicitly indicates that our method enables the model o learn meaningful features to overcome underfitting, even under strong data augmentation.

**Hyper-parameters.** Figure 6 plots the FID results of AugSelf-GAN with different hyper-parameters $\lambda = \lambda_d = \lambda_g$ ranging from $[0, 10]$ on CIFAR-10 and CIFAR-100. Notice that $\lambda = 0$ corresponds to the baseline BigGAN + DiffAugment. AugSelf-BigGAN performs the best when $\lambda$ is near 1. It is worth noting that AugSelf-BigGAN outperforms the baseline even for $\lambda = 10$ with 10% and 20% training data, demonstrating superior robustness with respect to the hyper-parameter $\lambda$.

## 7 Conclusion

This paper proposes a data-efficient GAN training method by utilizing augmentation parameters as self-supervision. Specifically, a novel self-supervised discriminator is proposed for predicting the augmentation parameters and data authenticity of augmented (real and generated) data simultaneously, given the original data. Meanwhile, the generator is encouraged to generate real rather than fake data of which augmentation parameters can be recognized by the self-supervised discriminator after augmentation. Theoretical analysis reveals a connection between the optimization objective of the generator and the arithmetic $-$ harmonic mean divergence under certain assumptions. Experiments on data-limited benchmarks demonstrate superior qualitative and quantitative performance of the proposed method compared to previous methods.

**Limitations.** In our experiments, we observed less significant improvement of AugSelf-GAN under sufficient training data. Furthermore, its effectiveness depends on the specific data augmentation used. In some cases, inappropriate data augmentation may limit the performance gain.

**Broader impacts.** This work aims at improving GANs under limited training data. While this may result in negative societal impacts, such as lowering the threshold of generating fake content or exacerbating bias and discrimination due to data issues, we believe that these risks can be mitigated. By establishing ethical guidelines for users and exploring fake content detection techniques, one can prevent these undesirable outcomes. Furthermore, this work contributes to the overall development of GANs and even generative models, ultimately promoting their potential benefits for society.

## Acknowledgements

We thank the anonymous reviewers for their valuable and constructive feedback. This work is funded by the National Natural Science Foundation of China under Grant Nos. 62272125, 62102402, U21B2046. Huawei Shen is also supported by Beijing Academy of Artificial Intelligence (BAAI).

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

# A    Proofs

**Proposition 1.** *For any generator $G$ and given unlimited capacity in the function space, the optimal augmentation-aware self-supervised discriminator $\hat{D}^*$ has the form of:*

$$\hat{D}^*(\hat{\mathbf{x}}, \mathbf{x}) = \frac{\int p_{\text{data}}(\mathbf{x}, \boldsymbol{\omega}, \hat{\mathbf{x}})\boldsymbol{\omega}^+ \mathrm{d}\boldsymbol{\omega} + \int p_G(\mathbf{x}, \boldsymbol{\omega}, \hat{\mathbf{x}})\boldsymbol{\omega}^- \mathrm{d}\boldsymbol{\omega}}{p_{\text{data}}(\mathbf{x}, \hat{\mathbf{x}}) + p_G(\mathbf{x}, \hat{\mathbf{x}})} \qquad (9)$$

*Proof.* The objective function of the self-supervised discriminator can be written as follows:

$$
\begin{aligned}
\mathcal{L}_{\hat{D}}^{\text{ss}} &= \mathbb{E}_{\mathbf{x}, \boldsymbol{\omega}}\left[\|\hat{D}(T(\mathbf{x}; \boldsymbol{\omega}), \mathbf{x}) - \boldsymbol{\omega}^+\|_2^2\right] + \mathbb{E}_{\mathbf{z}, \boldsymbol{\omega}}\left[\|\hat{D}(T(G(\mathbf{z}); \boldsymbol{\omega}), G(\mathbf{z})) - \boldsymbol{\omega}^-\|_2^2\right] \\
&= \iiint \left[p_{\text{data}}(\mathbf{x}, \boldsymbol{\omega}, \hat{\mathbf{x}})\|\hat{D}(\hat{\mathbf{x}}, \mathbf{x}) - \boldsymbol{\omega}^+\|_2^2 + p_G(\mathbf{x}, \boldsymbol{\omega}, \hat{\mathbf{x}})\|\hat{D}(\hat{\mathbf{x}}, \mathbf{x}) - \boldsymbol{\omega}^-\|_2^2\right]\mathrm{d}\mathbf{x}\mathrm{d}\boldsymbol{\omega}\mathrm{d}\hat{\mathbf{x}}.
\end{aligned}
$$

Minimizing the integral objective is equivalent to minimizing the objective on each data point:

$$\mathcal{L}_{\hat{D}}^{\text{ss}}(\mathbf{x}, \hat{\mathbf{x}}) = \int \left[p_{\text{data}}(\mathbf{x}, \boldsymbol{\omega}, \hat{\mathbf{x}})\|\hat{D}(\hat{\mathbf{x}}, \mathbf{x}) - \boldsymbol{\omega}^+\|_2^2 + p_G(\mathbf{x}, \boldsymbol{\omega}, \hat{\mathbf{x}})\|\hat{D}(\hat{\mathbf{x}}, \mathbf{x}) - \boldsymbol{\omega}^-\|_2^2\right]\mathrm{d}\boldsymbol{\omega}.$$

Getting its derivative with respect to the self-supervised discriminator and let it equals to 0, we have the optimal self-supervised discriminator as:

$$
\begin{aligned}
\frac{\partial \mathcal{L}_{\hat{D}}^{\text{ss}}}{\partial \hat{D}(\hat{\mathbf{x}}, \mathbf{x})} &= \int \left[p_{\text{data}}(\mathbf{x}, \boldsymbol{\omega}, \hat{\mathbf{x}})2(\hat{D}(\hat{\mathbf{x}}, \mathbf{x}) - \boldsymbol{\omega}^+) + p_G(\mathbf{x}, \boldsymbol{\omega}, \hat{\mathbf{x}})2(\hat{D}(\hat{\mathbf{x}}, \mathbf{x}) - \boldsymbol{\omega}^-)\right]\mathrm{d}\boldsymbol{\omega} = 0 \\
\Rightarrow \hat{D}^*(\hat{\mathbf{x}}, \mathbf{x}) &= \frac{\int p_{\text{data}}(\mathbf{x}, \boldsymbol{\omega}, \hat{\mathbf{x}})\boldsymbol{\omega}^+ \mathrm{d}\boldsymbol{\omega} + \int p_G(\mathbf{x}, \boldsymbol{\omega}, \hat{\mathbf{x}})\boldsymbol{\omega}^- \mathrm{d}\boldsymbol{\omega}}{p_{\text{data}}(\mathbf{x}, \hat{\mathbf{x}}) + p_G(\mathbf{x}, \hat{\mathbf{x}})}.
\end{aligned}
$$

$\square$

**Theorem 1.** *Assume that $\boldsymbol{\omega}^+ = -\boldsymbol{\omega}^- = \boldsymbol{c}$ is constant, under the optimal self-supervised discriminator $\hat{D}^*$, optimizing the self-supervised task for the generator $G$ is equivalent to:*

$$\min_G 4c \cdot M_{\text{AH}}(p_{\text{data}}(\mathbf{x}, \hat{\mathbf{x}})\|p_G(\mathbf{x}, \hat{\mathbf{x}})), \qquad (10)$$

*where $c = \|\boldsymbol{c}\|_2^2$ is constant and $M_{\text{AH}}$ is the arithmetic $-$ harmonic mean divergence [37], of which the minimum is achieved if and only if $p_G(\mathbf{x}, \hat{\mathbf{x}}) = p_{\text{data}}(\mathbf{x}, \hat{\mathbf{x}}) \Rightarrow p_G(\mathbf{x}) = p_{\text{data}}(\mathbf{x})$.*

*Proof.* The objective function of the self-supervised task for the generator can be written as follows:

$$
\begin{aligned}
&\mathbb{E}_{\mathbf{z}, \boldsymbol{\omega}}\left[\|\hat{D}(T(G(\mathbf{z}); \boldsymbol{\omega}), G(\mathbf{z})) - \boldsymbol{\omega}^+\|_2^2\right] - \mathbb{E}_{\mathbf{z}, \boldsymbol{\omega}}\left[\|\hat{D}(T(G(\mathbf{z}); \boldsymbol{\omega}), G(\mathbf{z})) - \boldsymbol{\omega}^-\|_2^2\right] \\
&= \iiint p_G(\mathbf{x}, \boldsymbol{\omega}, \hat{\mathbf{x}})\left[\|\hat{D}(\mathbf{x}, \hat{\mathbf{x}}) - \boldsymbol{\omega}^+\|_2^2 - \|\hat{D}(\mathbf{x}, \hat{\mathbf{x}}) - \boldsymbol{\omega}^-\|_2^2\right]\mathrm{d}\mathbf{x}\mathrm{d}\boldsymbol{\omega}\mathrm{d}\hat{\mathbf{x}} \\
&= \iint p_G(\mathbf{x}, \hat{\mathbf{x}})\left[\|\frac{\boldsymbol{c}(p_{\text{data}}(\mathbf{x}, \hat{\mathbf{x}}) - p_G(\mathbf{x}, \hat{\mathbf{x}}))}{p_{\text{data}}(\mathbf{x}, \hat{\mathbf{x}}) + p_G(\mathbf{x}, \hat{\mathbf{x}})} - \boldsymbol{c}\|_2^2 - \|\frac{\boldsymbol{c}(p_{\text{data}}(\mathbf{x}, \hat{\mathbf{x}}) - p_G(\mathbf{x}, \hat{\mathbf{x}}))}{p_{\text{data}}(\mathbf{x}, \hat{\mathbf{x}}) + p_G(\mathbf{x}, \hat{\mathbf{x}})} + \boldsymbol{c}\|_2^2\right]\mathrm{d}\mathbf{x}\mathrm{d}\hat{\mathbf{x}} \\
&= c \cdot \iint p_G(\mathbf{x}, \hat{\mathbf{x}})\left[\|\frac{p_{\text{data}}(\mathbf{x}, \hat{\mathbf{x}}) - p_G(\mathbf{x}, \hat{\mathbf{x}})}{p_{\text{data}}(\mathbf{x}, \hat{\mathbf{x}}) + p_G(\mathbf{x}, \hat{\mathbf{x}})} - 1\|_2^2 - \|\frac{p_{\text{data}}(\mathbf{x}, \hat{\mathbf{x}}) - p_G(\mathbf{x}, \hat{\mathbf{x}})}{p_{\text{data}}(\mathbf{x}, \hat{\mathbf{x}}) + p_G(\mathbf{x}, \hat{\mathbf{x}})} + 1\|_2^2\right]\mathrm{d}\mathbf{x}\mathrm{d}\hat{\mathbf{x}} \\
&= c \cdot \iint p_G(\mathbf{x}, \hat{\mathbf{x}})\left[\|\frac{2p_G(\mathbf{x}, \hat{\mathbf{x}})}{p_{\text{data}}(\mathbf{x}, \hat{\mathbf{x}}) + p_G(\mathbf{x}, \hat{\mathbf{x}})}\|_2^2 - \|\frac{2p_{\text{data}}(\mathbf{x}, \hat{\mathbf{x}})}{p_{\text{data}}(\mathbf{x}, \hat{\mathbf{x}}) + p_G(\mathbf{x}, \hat{\mathbf{x}})}\|_2^2\right]\mathrm{d}\mathbf{x}\mathrm{d}\hat{\mathbf{x}} \\
&= 4c \cdot \iint p_G(\mathbf{x}, \hat{\mathbf{x}})\left[\frac{p_G(\mathbf{x}, \hat{\mathbf{x}})^2 - p_{\text{data}}(\mathbf{x}, \hat{\mathbf{x}})^2}{(p_{\text{data}}(\mathbf{x}, \hat{\mathbf{x}}) + p_G(\mathbf{x}, \hat{\mathbf{x}}))^2}\right]\mathrm{d}\mathbf{x}\mathrm{d}\hat{\mathbf{x}} \\
&= 4c \cdot \iint p_G(\mathbf{x}, \hat{\mathbf{x}})\left[\frac{p_G(\mathbf{x}, \hat{\mathbf{x}}) - p_{\text{data}}(\mathbf{x}, \hat{\mathbf{x}})}{p_{\text{data}}(\mathbf{x}, \hat{\mathbf{x}}) + p_G(\mathbf{x}, \hat{\mathbf{x}})}\right]\mathrm{d}\mathbf{x}\mathrm{d}\hat{\mathbf{x}} \\
&= 4c \cdot M_{\text{AH}}(p_{\text{data}}(\mathbf{x}, \hat{\mathbf{x}})\|p_G(\mathbf{x}, \hat{\mathbf{x}})),
\end{aligned}
$$

where $c = \|\boldsymbol{c}\|_2^2$ is a constant scalar for the constant vector $\boldsymbol{c} = \boldsymbol{\omega}^+ = -\boldsymbol{\omega}^-$. And we have the optimal generator $p_G(\mathbf{x}, \hat{\mathbf{x}}) = p_{\text{data}}(\mathbf{x}, \hat{\mathbf{x}}) \Rightarrow p_G(\mathbf{x}) = p_{\text{data}}(\mathbf{x})$ to minimize the AHM divergence.    $\square$

Table 6: IS and FID of AugSelf-BigGAN with different self-supervised tasks (SS: Equations (19) and (21); SS+: Equations (20) and (21); ASS: Equations (5) and (6)) on CIFAR-10 and CIFAR-100 with full and limited training data. The best is **bold** and the second best is underlined.

| | Method | 100% training data | | 20% training data | | 10% training data | |
|---|---|---|---|---|---|---|---|
| | | IS ($\uparrow$) | FID ($\downarrow$) | IS ($\uparrow$) | FID ($\downarrow$) | IS ($\uparrow$) | FID ($\downarrow$) |
| CIFAR-10 | Baseline | $\underline{9.29}_{\pm 0.02}$ | $8.48_{\pm 0.13}$ | $8.84_{\pm 0.12}$ | $15.14_{\pm 0.47}$ | $\mathbf{8.80}_{\pm 0.01}$ | $20.60_{\pm 0.13}$ |
| | SS ($\lambda_g = 0$) | $9.28_{\pm 0.06}$ | $8.30_{\pm 0.12}$ | $\underline{8.86}_{\pm 0.09}$ | $13.96_{\pm 0.19}$ | $8.62_{\pm 0.09}$ | $20.94_{\pm 0.42}$ |
| | SS | $\underline{9.29}_{\pm 0.07}$ | $8.24_{\pm 0.10}$ | $\mathbf{8.98}_{\pm 0.04}$ | $\underline{13.42}_{\pm 0.36}$ | $8.69_{\pm 0.05}$ | $19.40_{\pm 0.28}$ |
| | SS+ ($\lambda_g = 0$) | $9.25_{\pm 0.06}$ | $\underline{8.11}_{\pm 0.18}$ | $8.81_{\pm 0.05}$ | $13.86_{\pm 0.31}$ | $\underline{8.76}_{\pm 0.06}$ | $19.52_{\pm 0.24}$ |
| | SS+ | $9.26_{\pm 0.04}$ | $8.26_{\pm 0.28}$ | $8.85_{\pm 0.05}$ | $13.81_{\pm 0.27}$ | $8.63_{\pm 0.05}$ | $19.85_{\pm 0.18}$ |
| | ASS ($\lambda_g = 0$) | $9.12_{\pm 0.05}$ | $8.90_{\pm 0.07}$ | $8.82_{\pm 0.03}$ | $13.65_{\pm 0.70}$ | $8.52_{\pm 0.04}$ | $\underline{18.14}_{\pm 0.33}$ |
| | ASS | $\mathbf{9.43}_{\pm 0.14}$ | $\mathbf{7.68}_{\pm 0.06}$ | $\mathbf{8.98}_{\pm 0.09}$ | $\mathbf{10.97}_{\pm 0.09}$ | $\underline{8.76}_{\pm 0.05}$ | $\mathbf{15.68}_{\pm 0.26}$ |
| CIFAR-100 | Baseline | $11.02_{\pm 0.07}$ | $11.49_{\pm 0.21}$ | $9.45_{\pm 0.05}$ | $24.98_{\pm 0.48}$ | $8.50_{\pm 0.09}$ | $34.92_{\pm 0.63}$ |
| | SS ($\lambda_g = 0$) | $\underline{11.04}_{\pm 0.09}$ | $11.48_{\pm 0.43}$ | $9.81_{\pm 0.07}$ | $21.17_{\pm 0.19}$ | $9.11_{\pm 0.09}$ | $30.48_{\pm 0.57}$ |
| | SS | $10.96_{\pm 0.11}$ | $11.04_{\pm 0.15}$ | $\underline{9.99}_{\pm 0.09}$ | $20.72_{\pm 0.33}$ | $9.23_{\pm 0.07}$ | $31.54_{\pm 1.12}$ |
| | SS+ ($\lambda_g = 0$) | $10.84_{\pm 0.09}$ | $11.48_{\pm 0.13}$ | $9.95_{\pm 0.07}$ | $21.22_{\pm 0.57}$ | $9.14_{\pm 0.05}$ | $31.02_{\pm 1.00}$ |
| | SS+ | $10.94_{\pm 0.10}$ | $\underline{10.94}_{\pm 0.12}$ | $9.88_{\pm 0.06}$ | $22.72_{\pm 0.37}$ | $9.23_{\pm 0.14}$ | $31.40_{\pm 0.22}$ |
| | ASS ($\lambda_g = 0$) | $10.82_{\pm 0.10}$ | $11.29_{\pm 0.12}$ | $9.96_{\pm 0.11}$ | $\underline{18.90}_{\pm 0.41}$ | $\underline{9.45}_{\pm 0.05}$ | $\underline{25.77}_{\pm 0.95}$ |
| | ASS | $\mathbf{11.19}_{\pm 0.09}$ | $\mathbf{9.88}_{\pm 0.07}$ | $\mathbf{10.25}_{\pm 0.06}$ | $\mathbf{16.11}_{\pm 0.25}$ | $9.78_{\pm 0.08}$ | $21.30_{\pm 0.15}$ |

**Corollary 1.** *The following equality and inequality hold for the AHM divergence:*

- $M_{\mathrm{AH}}(p_{\mathrm{data}}(\mathbf{x}, \hat{\mathbf{x}}) \| p_G(\mathbf{x}, \hat{\mathbf{x}})) + M_{\mathrm{AH}}(p_G(\mathbf{x}, \hat{\mathbf{x}}) \| p_{\mathrm{data}}(\mathbf{x}, \hat{\mathbf{x}})) = \Delta(p_{\mathrm{data}}(\mathbf{x}, \hat{\mathbf{x}}) \| p_G(\mathbf{x}, \hat{\mathbf{x}}))$
- $M_{\mathrm{AH}}(p_{\mathrm{data}}(\mathbf{x}, \hat{\mathbf{x}}) \| p_G(\mathbf{x}, \hat{\mathbf{x}})) = 1 - W(p_{\mathrm{data}}(\mathbf{x}, \hat{\mathbf{x}}) \| p_G(\mathbf{x}, \hat{\mathbf{x}})) \leq 1$

*where $\Delta$ is the Le Cam (LC) divergence [23], and $W$ is the harmonic mean divergence [37].*

*Proof.* We first prove the first corollary:

$$M_{\mathrm{AH}}(p_{\mathrm{data}}(\mathbf{x}, \hat{\mathbf{x}}) \| p_G(\mathbf{x}, \hat{\mathbf{x}}))$$
$$\leq M_{\mathrm{AH}}(p_{\mathrm{data}}(\mathbf{x}, \hat{\mathbf{x}}) \| p_G(\mathbf{x}, \hat{\mathbf{x}})) + M_{\mathrm{AH}}(p_G(\mathbf{x}, \hat{\mathbf{x}}) \| p_{\mathrm{data}}(\mathbf{x}, \hat{\mathbf{x}}))$$
$$= \iint p_G(\mathbf{x}, \hat{\mathbf{x}}) \frac{p_G(\mathbf{x}, \hat{\mathbf{x}}) - p_{\mathrm{data}}(\mathbf{x}, \hat{\mathbf{x}})}{p_{\mathrm{data}}(\mathbf{x}, \hat{\mathbf{x}}) + p_G(\mathbf{x}, \hat{\mathbf{x}})} \mathrm{d}\mathbf{x}\mathrm{d}\hat{\mathbf{x}} + \iint p_{\mathrm{data}}(\mathbf{x}, \hat{\mathbf{x}}) \frac{p_{\mathrm{data}}(\mathbf{x}, \hat{\mathbf{x}}) - p_G(\mathbf{x}, \hat{\mathbf{x}})}{p_G(\mathbf{x}, \hat{\mathbf{x}}) + p_{\mathrm{data}}(\mathbf{x}, \hat{\mathbf{x}})} \mathrm{d}\mathbf{x}\mathrm{d}\hat{\mathbf{x}}$$
$$= \iint p_G(\mathbf{x}, \hat{\mathbf{x}}) \frac{p_G(\mathbf{x}, \hat{\mathbf{x}}) - p_{\mathrm{data}}(\mathbf{x}, \hat{\mathbf{x}})}{p_{\mathrm{data}}(\mathbf{x}, \hat{\mathbf{x}}) + p_G(\mathbf{x}, \hat{\mathbf{x}})} \mathrm{d}\mathbf{x}\mathrm{d}\hat{\mathbf{x}} - \iint p_{\mathrm{data}}(\mathbf{x}, \hat{\mathbf{x}}) \frac{p_G(\mathbf{x}, \hat{\mathbf{x}}) - p_{\mathrm{data}}(\mathbf{x}, \hat{\mathbf{x}})}{p_G(\mathbf{x}, \hat{\mathbf{x}}) + p_{\mathrm{data}}(\mathbf{x}, \hat{\mathbf{x}})} \mathrm{d}\mathbf{x}\mathrm{d}\hat{\mathbf{x}}$$
$$= \iint \frac{(p_G(\mathbf{x}, \hat{\mathbf{x}}) - p_{\mathrm{data}}(\mathbf{x}, \hat{\mathbf{x}}))^2}{p_{\mathrm{data}}(\mathbf{x}, \hat{\mathbf{x}}) + p_G(\mathbf{x}, \hat{\mathbf{x}})} \mathrm{d}\mathbf{x}\mathrm{d}\hat{\mathbf{x}}$$
$$= \Delta(p_{\mathrm{data}}(\mathbf{x}, \hat{\mathbf{x}}) \| p_G(\mathbf{x}, \hat{\mathbf{x}})),$$

where $0 \leq \Delta(p_{\mathrm{data}}(\mathbf{x}, \hat{\mathbf{x}}) \| p_G(\mathbf{x}, \hat{\mathbf{x}})) \leq 2$ is the Le Cam (LC) divergence [23].

The following proves the second corollary:

$$0 \leq M_{\mathrm{AH}}(p_{\mathrm{data}}(\mathbf{x}, \hat{\mathbf{x}}) \| p_G(\mathbf{x}, \hat{\mathbf{x}}))$$
$$= \iint p_G(\mathbf{x}, \hat{\mathbf{x}}) \frac{p_G(\mathbf{x}, \hat{\mathbf{x}}) - p_{\mathrm{data}}(\mathbf{x}, \hat{\mathbf{x}})}{p_{\mathrm{data}}(\mathbf{x}, \hat{\mathbf{x}}) + p_G(\mathbf{x}, \hat{\mathbf{x}})} \mathrm{d}\mathbf{x}\mathrm{d}\hat{\mathbf{x}}$$
$$= \iint p_G(\mathbf{x}, \hat{\mathbf{x}}) \left[ 1 - \frac{2 p_{\mathrm{data}}(\mathbf{x}, \hat{\mathbf{x}})}{p_{\mathrm{data}}(\mathbf{x}, \hat{\mathbf{x}}) + p_G(\mathbf{x}, \hat{\mathbf{x}})} \right] \mathrm{d}\mathbf{x}\mathrm{d}\hat{\mathbf{x}}$$
$$= 1 - \iint \frac{2 p_{\mathrm{data}}(\mathbf{x}, \hat{\mathbf{x}}) p_G(\mathbf{x}, \hat{\mathbf{x}})}{p_{\mathrm{data}}(\mathbf{x}, \hat{\mathbf{x}}) + p_G(\mathbf{x}, \hat{\mathbf{x}})} \mathrm{d}\mathbf{x}\mathrm{d}\hat{\mathbf{x}}$$
$$= 1 - W(p_{\mathrm{data}}(\mathbf{x}, \hat{\mathbf{x}}) \| p_G(\mathbf{x}, \hat{\mathbf{x}})) \leq 1,$$

where $0 \leq W(p_{\mathrm{data}}(\mathbf{x}, \hat{\mathbf{x}}) \| p_G(\mathbf{x}, \hat{\mathbf{x}})) \leq 1$ is the well known harmonic mean divergence [37]. □

# B  Loss Functions

AugSelf-GAN adopts all kinds of augmentations of DiffAugment as the self-supervised signals by default, thus the self-supervised loss functions of the discriminator and the generator (Equations (5) and (6)) actually each comprise three sub self-supervised loss functions. Specifically, for AugSelf-GAN that uses color, translation, and cutout as self-supervision, the objective functions are:

$$\min_{D,\hat{D}_{\text{color}},\hat{D}_{\text{translation}},\hat{D}_{\text{cutout}}} \mathcal{L}_D^{\text{da}} + \lambda_d \cdot \left( \mathcal{L}_{\hat{D}_{\text{color}}}^{\text{color}} + \mathcal{L}_{\hat{D}_{\text{translation}}}^{\text{translation}} + \mathcal{L}_{\hat{D}_{\text{cutout}}}^{\text{cutout}} \right), \tag{11}$$

$$\min_G \mathcal{L}_G^{\text{da}} + \lambda_g \cdot \left( \mathcal{L}_G^{\text{color}} + \mathcal{L}_G^{\text{translation}} + \mathcal{L}_G^{\text{cutout}} \right), \tag{12}$$

where $\mathcal{L}_{\hat{D}_{\text{color}}}^{\text{color}}, \mathcal{L}_{\hat{D}_{\text{translation}}}^{\text{translation}}, \mathcal{L}_{\hat{D}_{\text{cutout}}}^{\text{cutout}}, \mathcal{L}_G^{\text{color}}, \mathcal{L}_G^{\text{translation}}$, and $\mathcal{L}_G^{\text{cutout}}$ are defined as:

$$\mathcal{L}_{\hat{D}_{\text{color}}}^{\text{color}} = \mathbb{E}_{\mathbf{x},\boldsymbol{\omega}} \left[ \|\hat{D}_{\text{color}}(T(\mathbf{x};\boldsymbol{\omega}),\mathbf{x}) - \boldsymbol{\omega}_{\text{color}}^+\|_2^2 \right]$$
$$+ \mathbb{E}_{\mathbf{z},\boldsymbol{\omega}} \left[ \|\hat{D}_{\text{color}}(T(G(\mathbf{z});\boldsymbol{\omega}),G(\mathbf{z})) - \boldsymbol{\omega}_{\text{color}}^-\|_2^2 \right], \tag{13}$$

$$\mathcal{L}_{\hat{D}_{\text{translation}}}^{\text{translation}} = \mathbb{E}_{\mathbf{x},\boldsymbol{\omega}} \left[ \|\hat{D}_{\text{translation}}(T(\mathbf{x};\boldsymbol{\omega}),\mathbf{x}) - \boldsymbol{\omega}_{\text{translation}}^+\|_2^2 \right]$$
$$+ \mathbb{E}_{\mathbf{z},\boldsymbol{\omega}} \left[ \|\hat{D}_{\text{translation}}(T(G(\mathbf{z});\boldsymbol{\omega}),G(\mathbf{z})) - \boldsymbol{\omega}_{\text{translation}}^-\|_2^2 \right], \tag{14}$$

$$\mathcal{L}_{\hat{D}_{\text{cutout}}}^{\text{cutout}} = \mathbb{E}_{\mathbf{x},\boldsymbol{\omega}} \left[ \|\hat{D}_{\text{cutout}}(T(\mathbf{x};\boldsymbol{\omega}),\mathbf{x}) - \boldsymbol{\omega}_{\text{cutout}}^+\|_2^2 \right]$$
$$+ \mathbb{E}_{\mathbf{z},\boldsymbol{\omega}} \left[ \|\hat{D}_{\text{cutout}}(T(G(\mathbf{z});\boldsymbol{\omega}),G(\mathbf{z})) - \boldsymbol{\omega}_{\text{cutout}}^-\|_2^2 \right], \tag{15}$$

$$\mathcal{L}_G^{\text{color}} = \mathbb{E}_{\mathbf{z},\boldsymbol{\omega}} \left[ \|\hat{D}_{\text{color}}(T(G(\mathbf{z});\boldsymbol{\omega}),G(\mathbf{z})) - \boldsymbol{\omega}_{\text{color}}^+\|_2^2 \right]$$
$$- \mathbb{E}_{\mathbf{z},\boldsymbol{\omega}} \left[ \|\hat{D}_{\text{color}}(T(G(\mathbf{z});\boldsymbol{\omega}),G(\mathbf{z})) - \boldsymbol{\omega}_{\text{color}}^-\|_2^2 \right], \tag{16}$$

$$\mathcal{L}_G^{\text{translation}} = \mathbb{E}_{\mathbf{z},\boldsymbol{\omega}} \left[ \|\hat{D}_{\text{translation}}(T(G(\mathbf{z});\boldsymbol{\omega}),G(\mathbf{z})) - \boldsymbol{\omega}_{\text{translation}}^+\|_2^2 \right]$$
$$- \mathbb{E}_{\mathbf{z},\boldsymbol{\omega}} \left[ \|\hat{D}_{\text{translation}}(T(G(\mathbf{z});\boldsymbol{\omega}),G(\mathbf{z})) - \boldsymbol{\omega}_{\text{translation}}^-\|_2^2 \right], \tag{17}$$

$$\mathcal{L}_G^{\text{cutout}} = \mathbb{E}_{\mathbf{z},\boldsymbol{\omega}} \left[ \|\hat{D}_{\text{cutout}}(T(G(\mathbf{z});\boldsymbol{\omega}),G(\mathbf{z})) - \boldsymbol{\omega}_{\text{cutout}}^+\|_2^2 \right]$$
$$- \mathbb{E}_{\mathbf{z},\boldsymbol{\omega}} \left[ \|\hat{D}_{\text{cutout}}(T(G(\mathbf{z});\boldsymbol{\omega}),G(\mathbf{z})) - \boldsymbol{\omega}_{\text{cutout}}^-\|_2^2 \right], \tag{18}$$

where $\hat{D}_{\text{color}} = \varphi_{\text{color}} \circ \phi$, $\hat{D}_{\text{translation}} = \varphi_{\text{translation}} \circ \phi$, and $\hat{D}_{\text{cutout}} = \varphi_{\text{cutout}} \circ \phi$ share the backbone $\phi$ but differ in the heads $\varphi_{\text{color}}, \varphi_{\text{translation}}$, or $\varphi_{\text{cutout}}$, respectively. For the simplicity of notations, we write Equations (5) and (6) in the main text as the objective function.

# C  Ablation Studies

**Self-supervised tasks.**  We introduce two non-adversarial self-supervised tasks for comparison. The first version is that the discriminator only learns self-supervision on real data, defined as:

$$\mathcal{L}_{\hat{D}}^{\text{ss}} = \mathbb{E}_{\mathbf{x}\sim p_{\text{data}}(\mathbf{x}),\boldsymbol{\omega}\sim p(\boldsymbol{\omega})} \left[ \|\hat{D}(T(\mathbf{x};\boldsymbol{\omega}),\mathbf{x}) - \boldsymbol{\omega}\|_2^2 \right]. \tag{19}$$

The second version is that the discriminator learns self-supervised tasks on both real and generated data simultaneously, given by:

$$\mathcal{L}_{\hat{D}}^{\text{ss}} = \mathbb{E}_{\mathbf{x},\boldsymbol{\omega}} \left[ \|\hat{D}(T(\mathbf{x};\boldsymbol{\omega}),\mathbf{x}) - \boldsymbol{\omega}\|_2^2 \right] + \mathbb{E}_{\mathbf{z},\boldsymbol{\omega}} \left[ \|\hat{D}(T(G(\mathbf{z});\boldsymbol{\omega}),G(\mathbf{z})) - \boldsymbol{\omega}\|_2^2 \right]. \tag{20}$$

For both versions, the generator is encouraged to produce augmentation-recognizable data, as follows:

$$\mathcal{L}_G^{\text{ss}} = \mathbb{E}_{\mathbf{z}\sim p(\mathbf{z}),\boldsymbol{\omega}\sim p(\boldsymbol{\omega})} \left[ \|\hat{D}(T(G(\mathbf{z});\boldsymbol{\omega}),G(\mathbf{z})) - \boldsymbol{\omega}\|_2^2 \right]. \tag{21}$$

According to the self-supervised task, we denote different approaches as SS (Equations (19) and (21)), SS+ (Equations (20) and (21)), and ASS (Equations (5) and (6), short for adversarial self-supervised learning, i.e., the proposed AugSelf-GAN).

Table 7: IS and FID of AugSelf-BigGAN with different self-supervised signals on CIFAR-10 and CIFAR-100 with full and limited training data. The self supervised signal to be predicted is marked by the symbol ✓. Notice that all methods adopt color, translation, and cutout as data augmentation.

| | color | trans. | cutout | IS ($\uparrow$) | FID ($\downarrow$) | IS ($\uparrow$) | FID ($\downarrow$) | IS ($\uparrow$) | FID ($\downarrow$) |
|---|---|---|---|---|---|---|---|---|---|
| | \multicolumn Self-Supervised Signals | | | 100% training data | | 20% training data | | 10% training data | |
| CIFAR-10 | ✗ | ✗ | ✗ | $9.29_{\pm.02}$ | $8.48_{\pm.13}$ | $8.84_{\pm.12}$ | $15.14_{\pm.47}$ | $8.80_{\pm.01}$ | $20.60_{\pm.13}$ |
| | ✗ | ✗ | ✓ | $9.30_{\pm.05}$ | $7.48_{\pm.07}$ | $8.94_{\pm.07}$ | $11.73_{\pm.27}$ | $8.73_{\pm.12}$ | $16.66_{\pm.39}$ |
| | ✗ | ✓ | ✗ | $9.39_{\pm.07}$ | $7.51_{\pm.12}$ | $8.95_{\pm.05}$ | $11.82_{\pm.21}$ | $8.80_{\pm.03}$ | $16.27_{\pm.35}$ |
| | ✗ | ✓ | ✓ | $9.38_{\pm.05}$ | $\mathbf{7.41}_{\pm.11}$ | $8.87_{\pm.04}$ | $11.19_{\pm.08}$ | $8.63_{\pm.08}$ | $16.30_{\pm.57}$ |
| | ✓ | ✗ | ✗ | $\mathbf{9.50}_{\pm.07}$ | $7.57_{\pm.07}$ | $\mathbf{8.99}_{\pm.06}$ | $11.38_{\pm.14}$ | $8.72_{\pm.09}$ | $16.50_{\pm.14}$ |
| | ✓ | ✗ | ✓ | $9.41_{\pm.07}$ | $7.51_{\pm.05}$ | $8.92_{\pm.12}$ | $11.20_{\pm.17}$ | $\mathbf{8.83}_{\pm.07}$ | $15.55_{\pm.21}$ |
| | ✓ | ✓ | ✗ | $9.42_{\pm.06}$ | $7.43_{\pm.06}$ | $8.91_{\pm.09}$ | $11.19_{\pm.20}$ | $8.58_{\pm.02}$ | $\mathbf{15.17}_{\pm.15}$ |
| | ✓ | ✓ | ✓ | $9.43_{\pm.14}$ | $7.68_{\pm.06}$ | $8.98_{\pm.09}$ | $\mathbf{10.97}_{\pm.09}$ | $8.76_{\pm.05}$ | $15.68_{\pm.26}$ |
| CIFAR-100 | ✗ | ✗ | ✗ | $11.02_{\pm.07}$ | $11.49_{\pm.21}$ | $9.45_{\pm.05}$ | $24.98_{\pm.48}$ | $8.50_{\pm.09}$ | $34.92_{\pm.63}$ |
| | ✗ | ✗ | ✓ | $11.16_{\pm.13}$ | $10.03_{\pm.14}$ | $10.18_{\pm.09}$ | $19.45_{\pm.62}$ | $9.46_{\pm.12}$ | $25.99_{\pm.62}$ |
| | ✗ | ✓ | ✗ | $\mathbf{11.35}_{\pm.10}$ | $9.88_{\pm.08}$ | $10.01_{\pm.07}$ | $18.39_{\pm.07}$ | $9.29_{\pm.10}$ | $26.50_{\pm.36}$ |
| | ✗ | ✓ | ✓ | $11.26_{\pm.07}$ | $9.75_{\pm.05}$ | $\mathbf{10.35}_{\pm.20}$ | $16.88_{\pm.46}$ | $9.92_{\pm.08}$ | $23.09_{\pm.72}$ |
| | ✓ | ✗ | ✗ | $11.20_{\pm.08}$ | $10.01_{\pm.13}$ | $9.90_{\pm.11}$ | $18.16_{\pm.62}$ | $9.39_{\pm.04}$ | $21.48_{\pm.14}$ |
| | ✓ | ✗ | ✓ | $11.17_{\pm.12}$ | $10.12_{\pm.20}$ | $10.14_{\pm.09}$ | $17.11_{\pm.62}$ | $\mathbf{9.94}_{\pm.12}$ | $24.52_{\pm.76}$ |
| | ✓ | ✓ | ✗ | $11.26_{\pm.16}$ | $\mathbf{9.65}_{\pm.04}$ | $10.21_{\pm.12}$ | $17.32_{\pm.51}$ | $9.92_{\pm.06}$ | $22.94_{\pm.17}$ |
| | ✓ | ✓ | ✓ | $11.19_{\pm.09}$ | $9.88_{\pm.07}$ | $10.25_{\pm.06}$ | $\mathbf{16.11}_{\pm.25}$ | $9.78_{\pm.08}$ | $\mathbf{21.30}_{\pm.15}$ |

Table 8: IS and FID of AugSelf-BigGAN with different architectures and fusions of $\varphi$ on CIFAR-10 and CIFAR-100 with 10% training data. The best result is **bold** and the second best is underlined. Input can be concatenation $[\phi(\mathbf{x}), \phi(\hat{\mathbf{x}})] \in \mathbb{R}^{2d}$, subtraction $\phi(\hat{\mathbf{x}}) - \phi(\mathbf{x})$, and augmentation $\phi(\hat{\mathbf{x}})$.

| Architecture | Input | CIFAR-10 10% data | | CIFAR-100 10% data | |
|---|---|---|---|---|---|
| | | IS ($\uparrow$) | FID ($\downarrow$) | IS ($\uparrow$) | FID ($\downarrow$) |
| two-layer MLP | concatenation | 8.54±0.07 | 18.47±0.37 | 8.65±0.11 | 30.15±0.47 |
| two-layer MLP | subtraction | 8.62±0.07 | 16.94±0.49 | 8.88±0.06 | 28.63±0.36 |
| two-layer MLP | augmentation | 8.78±0.05 | 19.06±0.81 | 8.73±0.10 | 29.40±0.91 |
| linear layer | concatenation | **8.85**±0.11 | 17.52±0.20 | 9.37±0.09 | 25.22±0.25 |
| linear layer | subtraction | 8.76±0.05 | **15.68**±0.26 | 9.78±0.08 | **21.30**±0.15 |
| linear layer | augmentation | 8.66±0.09 | 20.29±0.36 | **9.92**±0.10 | 24.47±0.58 |
| bilinear layer | - | 8.57±0.04 | 25.27±0.16 | 9.03±0.03 | 26.72±0.17 |

Table 6 reports the comparison between methods with different self-supervised tasks. We also conducted generator-free self-supervised learning experiments by setting the hyper-parameter as $\lambda_g = 0$ on each kind of self-supervised task. According to the FID score, ASS is significantly superior to SS and SS+. Even without the self-supervised task of the generator, ASS ($\lambda_g = 0$) outperforms SS and SS+ that include generator self-supervised tasks in limited (10% and 20%) data, and the introduction of generator self-supervised tasks further expands this advantage.

**Self-supervised signals.** We empirically analyze the role of different augmentation parameters as self-supervised signals for AugSelf-GAN. All methods employ three types of data augmentations (color, translation, and cutout), with the difference lying in the predicted self-supervised signals. According to the FID scores in Table 7, predicting any augmentation parameters can significantly improve the final generative performance compared to the baseline. Although there is no significant difference among them, we still choose to predict all augmentation parameters as the default setting for AugSelf-GAN. This experiment demonstrates that the self-supervised task itself plays a decisive role in training AugSelf-GAN, rather than the specific predicted augmentations. Therefore, we believe that our method is generalizable and can be extended to other advanced data augmentations.

**Network architectures.** We investigate the impact of different network architectures (two-layer multi-layer perceptron (MLP), linear layer, and bilinear layer) and input approaches (concatenation $[\phi(\mathbf{x}), \phi(\hat{\mathbf{x}})]$, subtraction $\phi(\hat{\mathbf{x}}) - \phi(\mathbf{x})$, and augmented samples only $\phi(\hat{\mathbf{x}})$) on AugSelf-GAN. As

Table 9: IS and FID of AugSelf-BigGAN with different loss functions on CIFAR-10 and CIFAR-100 with full and limited training data. The best result is **bold** and the second best is underlined.

| | Method | 100% training data | | 20% training data | | 10% training data | |
|---|---|---|---|---|---|---|---|
| | | IS ($\uparrow$) | FID ($\downarrow$) | IS ($\uparrow$) | FID ($\downarrow$) | IS ($\uparrow$) | FID ($\downarrow$) |
| CIFAR-10 | $\lambda_d = 0, \lambda_g = 0$ | $9.29_{\pm.02}$ | $8.48_{\pm.13}$ | $8.84_{\pm.12}$ | $15.14_{\pm.47}$ | $8.80_{\pm.01}$ | $20.60_{\pm.13}$ |
| | $\lambda_d = 1, \lambda_g = 0$ | $9.12_{\pm.05}$ | $8.90_{\pm.07}$ | $8.82_{\pm.03}$ | $13.65_{\pm.70}$ | $8.52_{\pm.04}$ | $18.14_{\pm.33}$ |
| | saturating | $9.27_{\pm.06}$ | $8.13_{\pm.14}$ | $8.83_{\pm.04}$ | $12.76_{\pm.29}$ | $8.42_{\pm.05}$ | $18.43_{\pm.21}$ |
| | non-saturating | $\underline{9.37}_{\pm.07}$ | $\mathbf{7.60}_{\pm.08}$ | $\underline{8.91}_{\pm.09}$ | $\underline{11.44}_{\pm.16}$ | $\underline{8.63}_{\pm.10}$ | $\underline{15.86}_{\pm.26}$ |
| | combination | $\mathbf{9.43}_{\pm.14}$ | $\underline{7.68}_{\pm.06}$ | $\mathbf{8.98}_{\pm.09}$ | $\mathbf{10.97}_{\pm.09}$ | $\mathbf{8.76}_{\pm.05}$ | $\mathbf{15.68}_{\pm.26}$ |
| CIFAR-100 | $\lambda_d = 0, \lambda_g = 0$ | $11.02_{\pm.07}$ | $11.49_{\pm.21}$ | $9.45_{\pm.05}$ | $24.98_{\pm.48}$ | $8.50_{\pm.09}$ | $34.92_{\pm.63}$ |
| | $\lambda_d = 1, \lambda_g = 0$ | $10.82_{\pm.10}$ | $11.29_{\pm.12}$ | $9.96_{\pm.11}$ | $18.90_{\pm.41}$ | $9.45_{\pm.05}$ | $25.77_{\pm.95}$ |
| | saturating | $10.90_{\pm.07}$ | $10.53_{\pm.21}$ | $9.71_{\pm.08}$ | $18.78_{\pm.43}$ | $9.30_{\pm.09}$ | $25.92_{\pm.12}$ |
| | non-saturating | $\mathbf{11.22}_{\pm.14}$ | $\underline{9.90}_{\pm.09}$ | $\underline{10.07}_{\pm.02}$ | $\underline{16.12}_{\pm.26}$ | $\mathbf{9.81}_{\pm.10}$ | $\underline{21.99}_{\pm.87}$ |
| | combination | $\underline{11.19}_{\pm.09}$ | $\mathbf{9.88}_{\pm.07}$ | $\mathbf{10.25}_{\pm.06}$ | $\mathbf{16.11}_{\pm.25}$ | $\underline{9.78}_{\pm.08}$ | $\mathbf{21.30}_{\pm.15}$ |

reported in Table 8, we found that more complicated architectures of the predict head $\varphi$ such as two-layer MLP with input concatenation ($\hat{D}(T(\mathbf{x}, \boldsymbol{\omega}), \mathbf{x}) = \mathtt{MLP}([\phi(T(\mathbf{x}, \boldsymbol{\omega})), \phi(\mathbf{x})])$) or bilinear layer ($\hat{D}(T(\mathbf{x}; \boldsymbol{\omega}), \mathbf{x}) = \phi(T(\mathbf{x}; \boldsymbol{\omega}))^\top W \phi(\mathbf{x}), W \in \mathbb{R}^{d \times d}$) works worse than the simple linear layer with input subtraction ($\hat{D}(T(\mathbf{x}; \boldsymbol{\omega}), \mathbf{x}) = \varphi(\phi(T(\mathbf{x}; \boldsymbol{\omega})) - \phi(\mathbf{x}))$), which is the design of AugSelf-GAN. The reason might be that complicated architectures of the head $\varphi$ actually discourage the backbone $\phi$ from capturing rich and linear representations, resulting in poor generalization of discriminators.

**Generator loss functions.** We study the effects of AugSelf-GANs with different generator loss functions. As shown in Table 9, The hyper-parameters $\lambda_d = 0, \lambda_g = 0$ represent the baseline BigGAN + DiffAugment. The generation performance is improved with only augmentation-aware self-supervision for the discriminator ($\lambda_d = 1, \lambda_g = 0$). The saturating version of self-supervised generator loss shows no significant further improvement. The non-saturating version yields comparable performance with the combination one (AugSelf-GAN), and they both substantially surpass the others. The reason is that the non-saturating loss directly encourages the generator to generate augmentation-predictable real data, providing more informative guidance than the saturating one.

# D   Additional Results

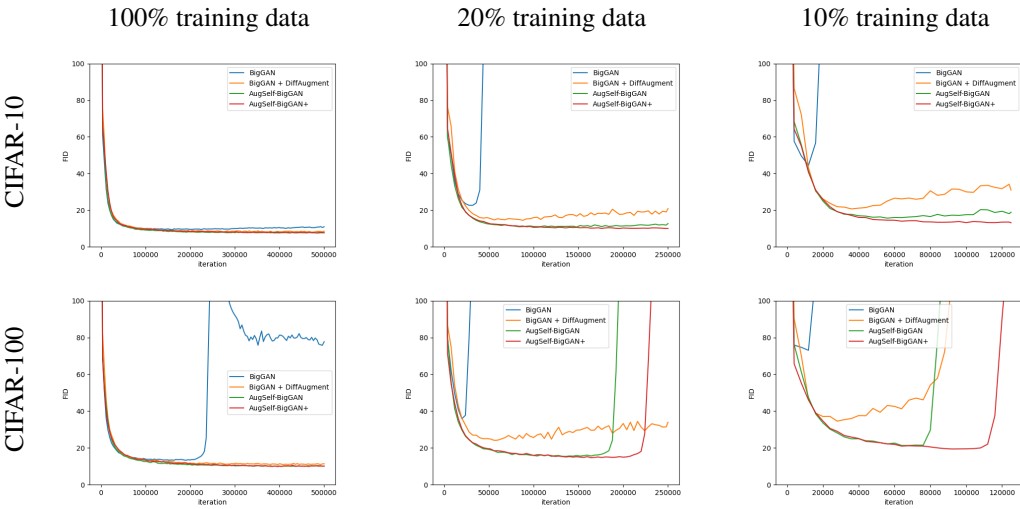

Figure 7: FID plots comparison of AugSelf-GAN and baselines on CIFAR-10 and CIFAR-100.

Table 10: FID comparison on AFHQ. The number within the parentheses represents the percentage of improvement ($\downarrow$) in AugSelf-StyleGAN2 compared to the baseline StyleGAN2 + DiffAugment.

| Method | Cat | Dog | Wild |
|---|---|---|---|
| StyleGAN2 [19] | 5.13 | 19.4 | 3.48 |
| + DiffAugment [47] | 3.49 | 8.75 | 2.69 |
| AugSelf-StyleGAN2 | **3.23** ($\downarrow$ 7.45%) | **8.17** ($\downarrow$ 6.63%) | **2.48** ($\downarrow$ 7.81%) |

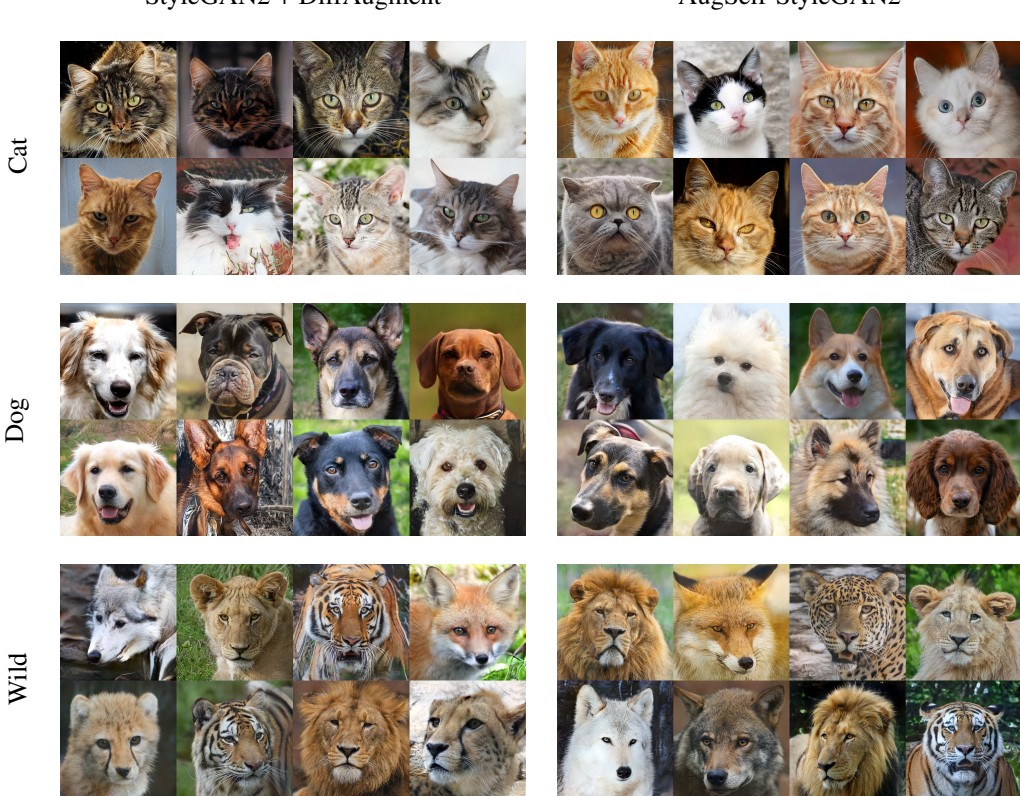

StyleGAN2 + DiffAugment           AugSelf-StyleGAN2

Figure 8: Visual comparison between StyleGAN2+DiffAugment and AugSelf-StyleGAN2 on AFHQ.

**Training stability.** Figure 7 shows the FID plots during training of our method and baselines on CIFAR-10 and CIFAR-100. Overall, AugSelf-BigGAN demonstrates a more stable convergence compared to the baselines, while AugSelf-BigGAN+ goes even further.

**AFHQ** We also conduct experiments on the AFHQ dataset [6] to compare our proposed method with DiffAugment by employing StyleGAN2 as the backbone model. The hyperparameters are set as $\lambda_d = 0.1$, $\lambda_d = 0.5$, and $\lambda_d = 0.2$ on Cat, Dog, and Wild domains, respectively, and $\lambda_g = 0.1$ on all three domains. Quantitative experimental results, as shown in Table 10, suggest that AugSelf-StyleGAN2 yields approximately a 7% improvements in FID across all three domains when compared to DiffAugment. Figure 8 presents a qualitative comparison between these two methods, demonstrating a significant improvement in image quality for our method.

**FFHQ and LSUN-Cat** Figures 9 and 10 present the randomly generated images produced by StyleGAN2 + DiffAugment and AugSelf-StyleGAN2+ on the FFHQ [17] and LSUN-Cat [45] datasets with varying amounts of training data, respectively. Our AugSelf-StyleGAN2+ demonstrates significant superiority over StyleGAN2 + DiffAugment in terms of visual quality of generated images.

StyleGAN2 + DiffAugment

AugSelf-StyleGAN2+

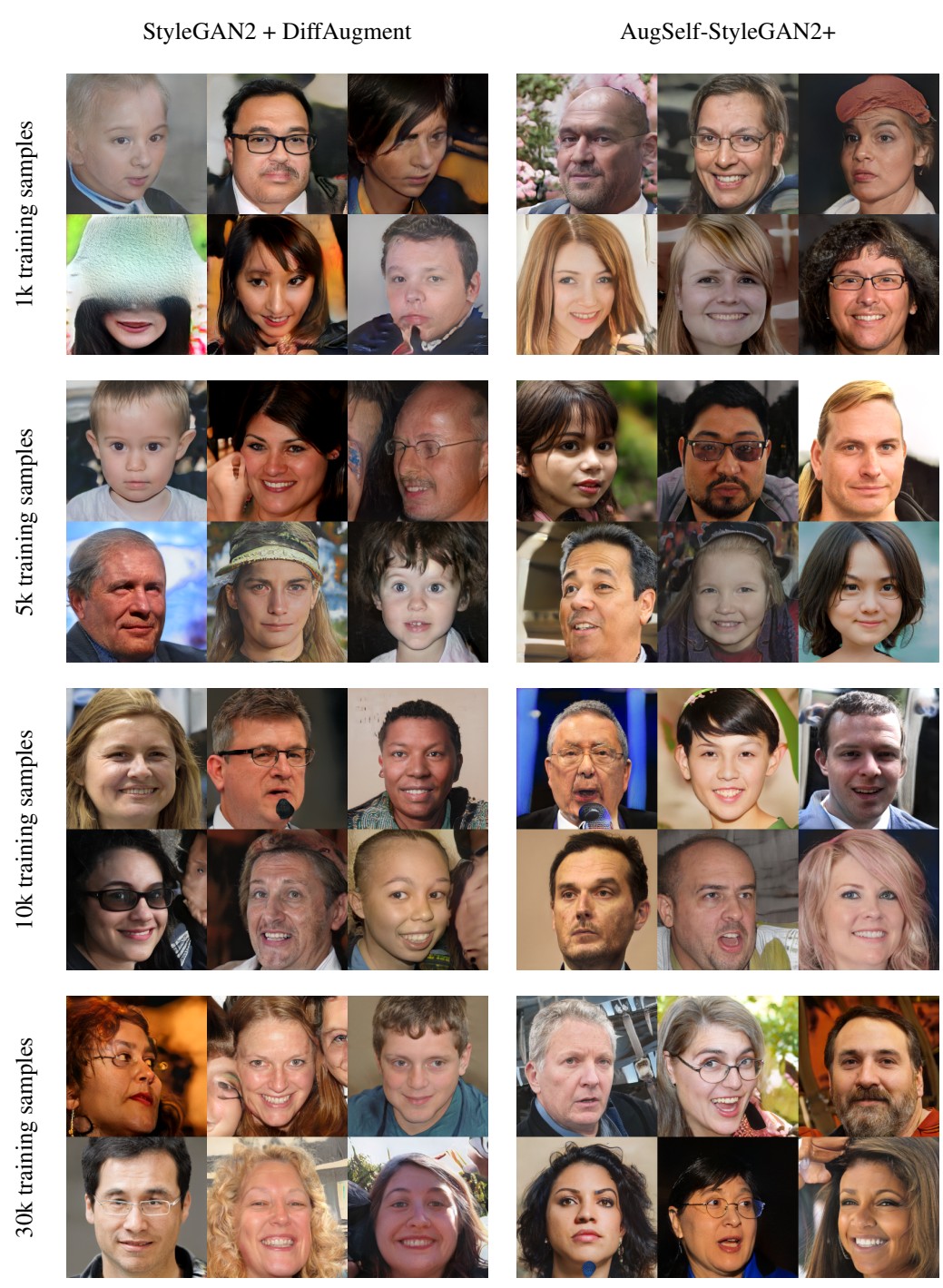

1k training samples

5k training samples

10k training samples

30k training samples

Figure 9: Comparison of random generated images between StyleGAN2 + DiffAugment and AugSelf-StyleGAN2+ on FFHQ with different amounts of training data.

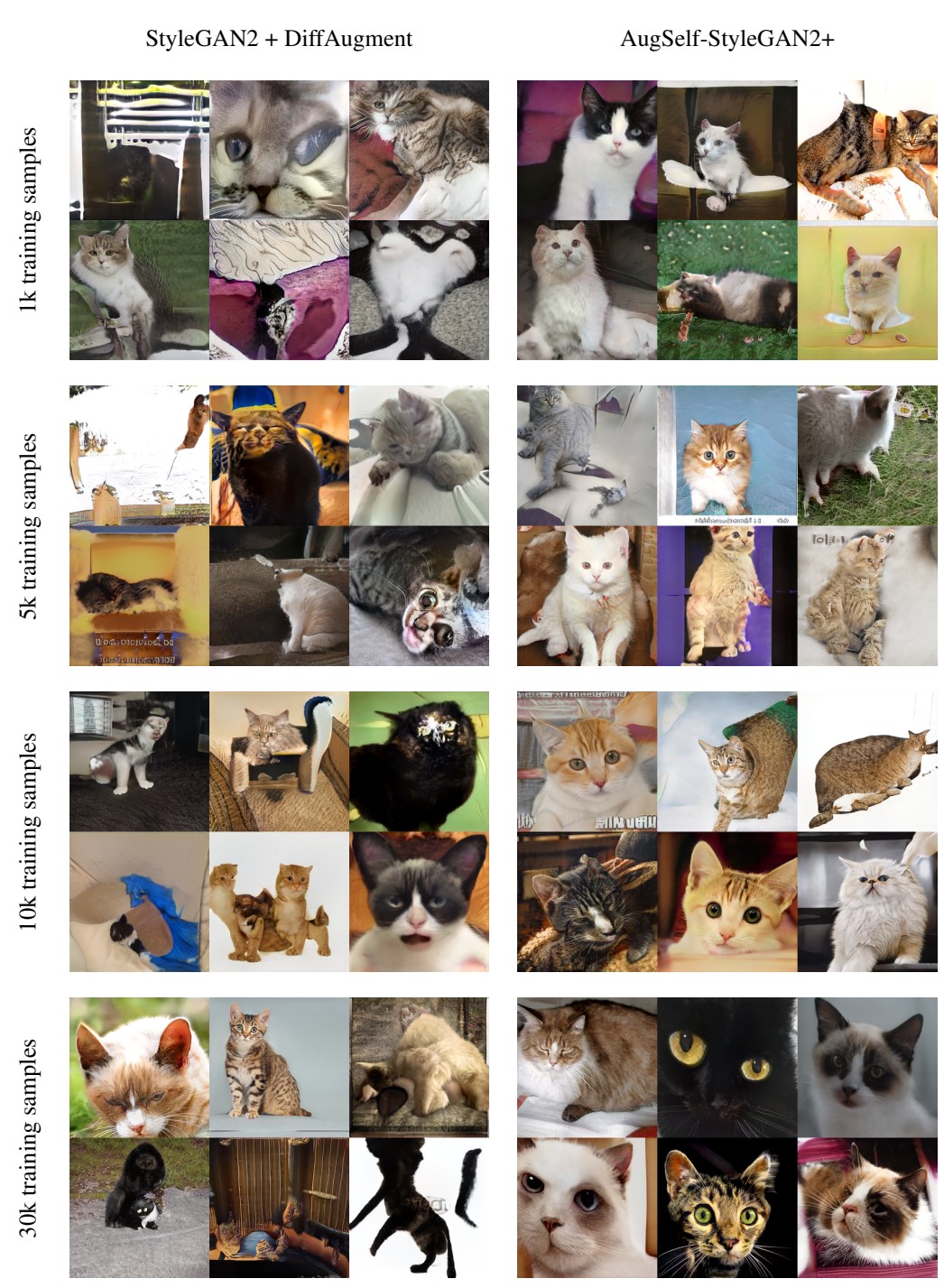

Figure 10: Comparison of randomly generated images between StyleGAN2 + DiffAugment and AugSelf-StyleGAN2+ on LSUN-Cat with different amounts of training data.

