# OpenReview forum: "Augmentation-Aware Self-Supervision for Data-Efficient GAN Training"
_NeurIPS.cc/2023/Conference — NeurIPS 2023 poster_

### Official Review · Reviewer_NBv7 · 2023-07-05

**Soundness:** 3 good
**Presentation:** 2 fair
**Contribution:** 2 fair
**Rating:** 5
**Confidence:** 5

**Summary:**

This paper introduces AugSelf-GAN, an enhanced approach for self-supervised GANs with label augmentation (SSGAN-LA). The authors present two major contributions. Firstly, they propose a discriminator that predicts continuous augmentation parameters, instead of classifying discrete augmentation methods. This allows the discriminator to learn a more fine-grained representation of the data distribution and to better distinguish between real and fake images. Secondly, the authors conduct a theoretical analysis that establishes a connection between the proposed method and the arithmetic-harmonic mean divergence. Experimental results demonstrate that AugSelf-GAN enhances GAN performance (as measured by FID and IS) in scenarios of limited data settings.

**Strengths:**

- The proposed method demonstrates significant improvements in quantitative results despite its simplicity.
- This paper offers comprehensive empirical and theoretical evidence to support the effectiveness of the proposed approach.

**Weaknesses:**

- L138–139: The authors claim that SSGAN-LA and ADA result in augmentation leaking. However, no evidence, such as citations or empirical results, is provided. I think this claim should be justified since it is the main motivation of the paper.

- The proposed method has limited novelty in terms of technical details. It can be seen as a variant of SSGAN-LA, with the main difference being that the discriminator predicts continuous augmentation parameters instead of discrete ones.

- The paper lacks a comparison with some previous methods, such as ContraD and SSGAN-LA. These methods are also designed to train GANs with data augmentation. It would be interesting (indeed necessary) to see how the proposed method compares to them.

- The authors argue that the representation learning capability of the discriminator could be improved by the proposed method. However, there are not enough experiments to support this claim. The experiments are conducted on CIFAR10/100 only (Fig. 2), and there are no comparisons with the methods that claim similar advantages (i.e., ContraD).

**Questions:**

Please see the above weakness.

**Limitations:**

Yes, the authors have discussed the limitations of the proposed method, as well as the potential negative impact of this work.

---

> ### Author Rebuttal · Authors · 2023-08-09
>
> We would like to thank Reviewer NBv7 for the valuable feedback. The comments and suggestions have greatly helped us in improving the quality of our work and ensuring its scientific rigor.
>
> ---
>
> **Augmentation leaking in SSGAN-LA and ADA?**
>
> In L138-139, we did not aim to claim that SSGAN-LA and ADA suffer from the problem of augmentation leaking; instead, we cited them to demonstrate how straightforward data augmentation may have this issue. We will clarify this statement more precisely in the updated version.
>
> ---
>
> **The proposed method has limited novelty in terms of technical details.**
>
> There are significant differences in the proposed methods and theoretical analysis between AugSelf-GAN and SSGAN-LA. SSGAN-LA employs a discrete adversarial self-supervised task, while AugSelf-GAN utilizes a continuous adversarial self-supervised task. It should be noted that the transition from discrete to continuous self-supervised adversarial tasks is not easy. Specifically, the adversarial idea requires us to distinguish self-supervised labels of real and generated data. In the discrete case, SSGAN-LA uses label augmentation to distinguish the class-label of real and fake data, while in the continuous case, there is no similar concept. Therefore, we must design a brand-new adversarial task that conforms to the continuous self-supervision. There may be multiple schemes for distinguishing continuous labels of real and fake data, but we innovatively choose the origin-symmetric approach and provide corresponding theoretical explanations. This theoretical explanation reveals the connection between the proposed method and a robust divergence, the arithmetic-harmonic mean (AHM) divergence, explaining the superiority of the method under limited data. This advantage is not present in SSGAN-LA, whose theoretically optimized divergence is the reversed KL divergence that is inferior to the AHM divergence in the studied scenario. Additionally, in terms of the specific implementation of the discriminator, SSGAN-LA does not accept original data as input and thus requires the data augmentation to be reversible, while AugSelf-GAN accepts both original and augmented data as input, making it unrestricted in data augmentation. We believe that the significant differences in method and theoretical analysis demonstrate the technical novelty of this work.
>
> ---
>
> **The paper lacks a comparison with some previous methods, such as ContraD and SSGAN-LA.**
>
> The following Table reports the comparison of AugSelf-GAN(+) with ContraD and SSGAN-LA on CIFAR-10 and CIFAR-100, showing that AugSelf-GAN(+) can outperform the best in limited-data settings in terms of FID, and we will also report it in the updated version of our paper.
>
> | IS, FID | CIFAR-10 | CIFAR-10-0.2 | CIFAR-10-0.1 | CIFAR-100 | CIFAR-100-0.2 | CIFAR-100-0.1 |
> | --- | --- | --- | --- | --- | --- | --- |
> | ContraD + StyleGAN2 | **9.47**, 9.80 | - | - | 10.00, 14.10 | - | - |
> | SSGAN-LA | 9.30, 8.05 | 8.84, 11.16 | 8.30, 15.08 | 11.13, **9.77** | **10.52**, 15.91 | 9.81, 23.17 |
> | AugSelf-GAN | 9.43, 7.68 | 8.98, 10.97 | 8.76, 15.68 | **11.19**, 9.88 | 10.25, 16.11 | 9.78, 21.30 |
> | AugSelf-GAN+ | 9.27, **7.54** | **9.08**, **9.95** | **8.79**, **12.76** | 11.12, 10.09 | 10.14, **15.33** | **9.93**, **18.64** |
>
> ---
>
> T**here are not enough experiments to support the improved representation learning capability of the discriminator of the proposed method.**
>
> Conducting representation learning experiments requires datasets with labels. In our experiments, CIFAR-10 and CIFAR-100 were the only datasets with labels available. We appreciate your valuable suggestions and have added Tiny ImageNet experiments. The experimental results are as follows, which also verify that this method can effectively improve the representation learning ability of the discriminator. We also report the IS and FID results, and will report the results in the updated version of our paper.
>
> | Acc | TI | TI-0.2 | TI-0.1 |
> | --- | --- | --- | --- |
> | BigGAN + DiffAugment | 0.3244 | 0.3073 | 0.3006 |
> | AugSelf-BigGAN | **0.4171** | **0.3577** | **0.3293** |
>
> | IS | TI | TI-0.2 | TI-0.1 |
> | --- | --- | --- | --- |
> | BigGAN + DiffAugment | 16.23+-0.22 | 13.83+-0.19 | 10.18+-0.08 |
> | AugSelf-BigGAN | **16.51+-0.19** | **14.81+-0.06** | **12.78+-0.19** |
>
> | FID | TI | TI-0.2 | TI-0.1 |
> | --- | --- | --- | --- |
> | BigGAN + DiffAugment | 17.11+-0.04 | 27.85+-0.16 | 41.96+-0.14 |
> | AugSelf-BigGAN | **15.82+-0.23** | **21.61+-0.27** | **29.47+-0.03** |
>
> ---
>
> **There are no comparisons with the methods that claim similar advantages.**
>
> We did not compare to ContraD in the submitted version as ContraD does not address the issue of data-limited GAN training, which is our studied problem, and it does not use BigGAN as the backbone on CIFAR-10/100, unlike the setting we followed with DiffAugment. Conducting code migration and experiments for this comparison would require significant time and resource, and we apologize that we could not provide comparison results in this short rebuttal phase. However, we will make the effort to migrate the code and conduct experiments to provide comparison results in the updated version of our paper.

---

> > ### Comment · Reviewer_NBv7 · 2023-08-19
> >
> > Thanks for the response and additional experiments.
> >
> > - Comparisons with SSGAN-LA
> >
> > We recognize the difference between SSGAN-LA and the proposed method. BTW, why "the transition from discrete to continuous self-supervised adversarial tasks" is necessarily more critical? Any implications?
> >
> > Besides, SSGAN-LA occasionally performs better than the proposed AugSelf-GAN or AugSelf-GAN++. Does the authors have any explanations or reasoning for performances?
> >
> > - Comparison with ContraD
> >
> > We gracefully disagree that ContraD does not target a limited data setting. It developed a data augmentation technique for GAN, thus it is naturally effective in a limited data scenario. The paper also stated that “We evaluate our training method on the Animal Faces-HQ (AFHQ) dataset (Choi et al., 2020), which consists of ∼15,000 samples of animal-face images at 512×512 resolution, to further verify the effectiveness of ContraD on higher-resolution, yet limited-sized datasets.”
> > (Besides, if the proposed method specifically targeted for a limited data setting does not outperform a general recipe, it is rather discouraging news. )
> >
> > Regarding the limited time issue, the data this paper dealing with is "CIFAR10/CIFAR100/Tiny-imagenet". It is not that heavy to conduct the GAN training on these datasets, given the rebuttal period.
> >
> >
> > Overall, we also read other reviewers' feedback and the authors' responses. Despite the difference of the proposed method from previous arts, we are still not convinced by the advantages given by the current experimental results.

---

> > > ### Author Response · Authors · 2023-08-21
> > > **Response by Authors**
> > >
> > > Dear Reviewer NBv7,
> > >
> > > Thank you for recognize the difference between SSGAN-LA and our proposed method.
> > >
> > > **Why "the transition from discrete to continuous self-supervised adversarial tasks" is necessarily**
> > >
> > > Our Theorem 1 links a relationship between AugSelf-GAN and the arithmetic-harmonic mean (AHM) divergence. The AHM divergence is smoother than the reverse KL divergence optimized by SSGAN-LA (Fig.4), which theoretically makes the training of AugSelf-GAN more stable in the limited data scenario, as it receives more robust feedback for the generator [1] for extremely large inputs ($p(x)/q(x)=D(x)/(1-D(x))$), which is estimated by discriminators and common in data-limited regimes where the discriminator's estimation of the proportion may be inaccurate. The essential reason behind this is: there is no size relationship between discrete labels, while continuous labels have a size relationship, so that it can control the strength of the discriminator to distinguish between real and fake ( in Eq.10), and alleviate its overfitting.
> > >
> > > **SSGAN-LA occasionally performs better than AugSelf-GAN(+)**
> > >
> > > We understand that SSGAN-LA outperforms AugSelf-GAN(+) on CIFAR-100 in terms of FID and on CIFAR-100 20% in terms of IS. For the first case, CIFAR-100 is a relatively data-rich dataset regarding the resolution, which may not play to the strengths of AugSelf-GAN(+). For the second case, it is being speculated that IS may not be an accurate metric for measuring the quality of the generated images, due to its inability to measure inter-class diversity [2]. This also contradicts the results of FID, where AugSelf GAN+ is superior to SSGAN-LA in this data setting.
> > >
> > > **Compare with ContraD on AFHQ**
> > >
> > > We appreciate the reviewer's suggestion to compare with ContraD on the AFHQ dataset and acknowledge that it would be a suitable dataset for evaluating our method. We are currently conducting experiments on the AFHQ dataset, but training on a single GPU for 512x512 resolution data takes at least 21 days and 22 hours to finish. We have obtained some promising results on AFHQ cat so far and will report the complete results in our updated version.
> > >
> > > |  | AFHQ cat |
> > > | --- | --- |
> > > | DiffAugment | 3.49 |
> > > | ContraD | 3.82 |
> > > | AugSelf-GAN | **3.37** |
> > >
> > > Regarding the claim that training on Tiny-ImageNet is not heavy, we respectfully disagree. Training on the entire Tiny-ImageNet dataset requires at least 5 days and 12 hours on a single GPU to complete 1000 epochs, and conducting multiple experiments is a challenge for our computing resources.
> > >
> > > Thank you again for your valuable feedback. We sincerely hope that our response can address your concerns.
> > >
> > > [1] Tseng, Hung-Yu, et al. Regularizing Generative Adversarial Networks under Limited Data. CVPR 2021
> > >
> > > [2] Heusel, Martin, et al. GANs Trained by a Two Time-scale Update Rule Converge to a Local Nash Equilibrium. NIPS 2017

---

> > > > ### Comment · Reviewer_NBv7 · 2023-08-21
> > > >
> > > > Thanks for the additional experiments and explanations.
> > > >
> > > > After reading all your supporting arguments and materials, I can now accept the advantage of the proposed method, the robust scheme given by the large inputs.
> > > >
> > > > We strongly suggest adding empirical evidence on various datasets in your final version because the usefulness of the training strategy indeed needs empirical verification.
> > > >
> > > > Considering all, as the authors properly answer my questions and concerns, I will increase my rating, to borderline accept.

---

### Official Review · Reviewer_nM6o · 2023-07-07

**Soundness:** 3 good
**Presentation:** 3 good
**Contribution:** 3 good
**Rating:** 5
**Confidence:** 4

**Summary:**

This paper proposes using a discriminator to predict the augmentation applied to the following sample, in addition to the standard real/fake discrimination objective. The authors demonstrate that using this objective training the GAN leads to minimizing to the Arithmetic-Harmonic Mean divergence objective. Further, the method shows improved results through FID in various data efficient setups of CIFAR-10, CIFAR-100, Few-Shot datasets etc.

**Strengths:**

1)	The proposed objective of predicting the augmentation through discriminator is novel to my knowledge.
2)	The paper is well written and clear.
3)	The technique of self-supervision of predicting the augmentation type seems to show significant improvement across baselines.

**Weaknesses:**

1) Reasoning for Effectiveness of Proposed Method: The authors argue that discriminator being invariant to augmentations is wrong. However, that seems intuitively sensible however the theory presented does not properly justify the reasoning for it’s effectiveness. A possible direction for this to think upon could be how the discriminator distribution expands to augmentation.

2) Convergence Plot: As the method seems to improve performance, an analysis of convergence of FID could be provided for clearing whether the proposed method delays collapse or prevents it all together. In L136 the authors mentioned about better convergence but do not provide FID plot.

Minor:
1) Fig. 2 is not described in text.
2) L136 misses citation.
3) Rephrase L49-51, as it's unclear.

**Questions:**

1) Missing Details: The major missing detail is about how to determine the value of w for various augmentations introduced in the paper. What is the general principle which is followed to determine this? This is also one of the major weaknesses, if there is no general way.

2) Missing Evaluation using CLIP FID. Recently it has been argued in multiple papers [R1,R2] that CLIP based FID is a better metric for evaluating GANS. Hence, I request authors to provide the same in rebuttal.

[R1]: The Role of ImageNet Classes in Fr\'echet Inception Distance
[R2]: NoisyTwins: Class-consistent and Diverse Image Generation through StyleGANs


Post Rebuttal Update: The authors responses were helpful and some of my major concerns have been resolved. I will maintain my borderline accept rating.

**Limitations:**

Yes, the authors have discussed limitations

---

> ### Author Rebuttal · Authors · 2023-08-09
>
> We would like to thank Reviewer nM6o for providing valuable feedback. The comments and suggestions have greatly helped us in improving the quality of our work and ensuring its scientific rigor.
>
> ---
>
> **Reasoning for Effectiveness of Proposed Method**
>
> We provided evidence of limitations to the representation learning of the discriminator of DiffAugment in Table 2, which motivated us to propose self-supervised task aiming to enhance this capability, and our experimental results have shown its effectiveness. We will modify the description of the existing limitations of DiffAugment to emphasizing that the discriminator lacks perception to different data augmentations, which limits the representation learning capability. Additionally, our theoretical analysis shows the robustness and superiority of the loss function of AugSelf-GAN in limited data settings.
>
> ---
>
> **Convergence Plot of FID.**
>
> We appreciate your constructive suggestion and have uploaded an extra PDF file that includes the FID convergence plots for AugSelf-BigGAN(+), DiffAugment, and BigGAN on CIFAR-10 and CIFAR-100 under different training data regimes for better analysis of convergence and performance comparison. As shown in Figure 11 (a,b,c,e,f,g), AugSelf-BigGAN(+) is significantly superior to the baselines DiffAugment and BigGAN on the FID convergence curve.
>
> ---
>
> **Fig. 2 is not described in text.**
>
> Fig. 2 is intended to demonstrate the representation learning capability of the discriminator. Specifically, we used the penultimate layer of the discriminator as a feature extractor and trained a downstream linear classifier on training set to classify the extracted feature on the validation set. We will include a corresponding description in the main text of the updated version.
>
> ---
>
> **L136 misses citation.**
>
> We will add citations to SSGAN [1] and SSGAN-LA [2] in L136 of the updated version to help clarify the motivation and impact of the experiment.
>
> ---
>
> **Rephrase L49-51, as it's unclear.**
>
> We will modify the description of L49-51 to make it more precise. The sentence aims to explain Equation 6, where the generated data from the generator through augmentation needs to be accurately identified by the self-supervised discriminator as real augmented data, instead of fake augmented data. We will strengthen the clarity of this statement by providing a reference to Equation 6 in the updated version.
>
> ---
>
> **Missing detail is about how to determine the value of w for various augmentations. What is the general principle which is followed to determine this?**
>
> In general, when applying data augmentation, the augmentation parameter $\omega$ follows a certain range $\Omega \in \mathbb{R}^d$ which is randomly sampled. In our work, we simply rescale the $\Omega$ space to $[0,1]^d$ to obtain the rescaled values of $\omega$. In our early experiments, we found that modifying the range of the rescaled space had little effect on the results, therefore we adopted the current approach to ensure the usability and generality of our method. We will clarify this in the updated version of our paper.
>
> ---
>
> **Missing Evaluation using CLIP FID.**
>
> Thank you for your valuable suggestion. We have followed your advice and evaluated augself(+) and diffaugment, using CLIP-FID on FFHQ and low-shot image generation. The experimental results, as shown in the table below, demonstrate that augself(+) outperforms diffaugment on most of the datasets.
>
> | CLIP-FID | FFHQ-30k | FFHQ-10k | FFHQ-5k | FFHQ-1k | 100-shot-obama | 100-shot-grumpy_cat | 100-shot-panda | AnimalFace-cat | AnimalFace-dog |
> | --- | --- | --- | --- | --- | --- | --- | --- | --- | --- |
> | DiffAugment | 3.67 | 5.64 | 7.53 | 17.47 | 15.98 | 16.04 | 15.46 | 22.68 | **22.69** |
> | AugSelf | **3.61** | 5.31 | 7.08 | **14.48** | **9.54** | **10.19** | **11.84** | **19.68** | 25.95 |
> | AugSelf+ | 4.01 | **4.87** | **7.03** | **14.48** |  |  |  |  |  |
>
> ---
>
> [1] Chen, et al. Self-Supervised GANs via Auxiliary Rotation Loss. CVPR 2019
>
> [2] Hou, et al. Self-Supervised GANs with Label Augmentation. NeurIPS 2021

---

> > ### Comment · Reviewer_nM6o · 2023-08-13
> > **Thanks for the Rebuttal**
> >
> > Dear authors,
> >
> > I have read the rebuttal and reviews from the other reviewers. I also find that current work closely relates to SSGAN-LA, which predicts discrete labels. I checked the response. However, I am not very convinced why the continuous task should provide improved results over discrete task.
> >
> > Also, can you provide concrete examples of how the $w$ parameter is defined for various augmentations?
> >
> > Thanks

---

> > > ### Author Response · Authors · 2023-08-14
> > > **Thanks for the Reply (1/2)**
> > >
> > > Dear Reviewer nM6o,
> > >
> > > Thank you very much for your reply. We would like to elaborate further to answer your question.
> > >
> > > ---
> > >
> > > - **The current work closely relates to SSGAN-LA, which predicts discrete labels. Why the continuous task should provide improved results over discrete task.**
> > >
> > > We want to first emphasize that AugSelf-GAN and SSGAN-LA have substantially distinct contributions despite sharing similar methodological ideas.
> > >
> > > Firstly, SSGAN-LA aims to address the biased learning objective issue in existing discrete transformation-based self-supervised GANs [1] by improving the self-supervised loss function [2]; whereas the goal of our AugSelf-GAN is to enhance GAN training with limited data by proposing a seamless-adaptive continuous adversarial self-supervised task based on DiffAugment.
> > >
> > > In terms of methodology, SSGAN-LA employs a discrete adversarial self-supervised task (classification), while AugSelf-GAN utilizes a continuous adversarial self-supervised task (regression). It should be noted that the transition from discrete to continuous self-supervised adversarial task is not straightforward. Specifically, the adversarial idea requires us to differentiate self-supervised labels between real and fake data. In the discrete case, the classification task makes it natural for SSGAN-LA to use label augmentation to differentiate self-supervised labels between real and fake data. However, continuous labels are generally predicted through regression which cannot utilize the label augmentation to differentiate self-supervised labels. AugSelf-GAN needs to design a novel continuous adversarial task that fits continuous labels. However, there are many ways to differentiate continuous labels, such as adding extra dimensions, adding offsets to the labels, etc., but not all of them can fit the requirement of generating modeling (fitting real data distribution). We innovatively choose the origin-symmetry scheme to differentiate self-supervised labels between real and fake data for AugSelf-GAN and provide theoretical explanations to show that the origin-symmetry scheme can fit the requirements of generating modeling under certain conditions.
> > >
> > > Moreover, with regard to the specific implementation of the discriminator, SSGAN-LA does not accept the raw data as input, and thus requires data augmentation to be reversible, otherwise it cannot learn the real data distribution, while AugSelf-GAN accepts both raw data and augmented data as input and is not subject to this requirement.
> > >
> > > The most important thing is that our theoretical analysis reveals **why the continuous task should provide improved results over discrete task in data-limited GAN training**. We link a relationship between AugSelf-GAN and the arithmetic-harmonic mean (AHM) divergence. The AHM divergence is smoother than the reverse KL divergence optimized by SSGAN-LA (Fig.4), which theoretically makes the training of AugSelf-GAN more stable in the limited data scenario, as it receives more robust feedback [3]. The essential reason behind this is: there is no size relationship between discrete labels, while continuous labels have a size relationship, so that it can control the strength of the discriminator to distinguish between real and fake ($c$ in Eq.10), and alleviate its overfitting.
> > >
> > > The Table below compares AugSelf-GAN and SSGAN-LA on the CIFAR-10 and CIFAR-100 with different percentages of training data. According to the FID scores, AugSelf-GAN outperforms SSGAN-LA in most settings. We will add this comparative experiment in the revised version of our paper.
> > >
> > > | FID | CIFAR-10 | CIFAR-10-0.2 | CIFAR-10-0.1 | CIFAR-100 | CIFAR-100-0.2 | CIFAR-100-0.1 |
> > > | --- | --- | --- | --- | --- | --- | --- |
> > > | SSGAN-LA | 8.05 | 11.16 | 15.08 | **9.77** | 15.91 | 23.17 |
> > > | AugSelf-GAN | 7.68 | 10.97 | 15.68 | 9.88 | 16.11 | 21.30 |
> > > | AugSelf-GAN+ | **7.54** | **9.95** | **12.76** | 10.09 | **15.33** | **18.64** |
> > >
> > > | IS | CIFAR-10 | CIFAR-10-0.2 | CIFAR-10-0.1 | CIFAR-100 | CIFAR-100-0.2 | CIFAR-100-0.1 |
> > > | --- | --- | --- | --- | --- | --- | --- |
> > > | SSGAN-LA | 9.30 | 8.84 | 8.30 | 11.13 | **10.52** | 9.81 |
> > > | AugSelf-GAN | **9.43** | 8.98 | 8.76 | **11.19** | 10.25 | 9.78 |
> > > | AugSelf-GAN+ | 9.27 | **9.08** | **8.79** | 11.12 | 10.14 | **9.93** |
> > >
> > > We believe that the above significant differences demonstrate the novelty and contribution of this work compared to SSGAN-LA.
> > >
> > > ---
> > >
> > > [1] Tran, Ngoc-Trung, et al. Self-supervised GAN: Analysis and Improvement with Multi-class Minimax Game. NeurIPS 2019.
> > >
> > > [2] Chen, Ting, et al. Self-Supervised GANs via Auxiliary Rotation Loss. CVPR 2019.
> > >
> > > [3] Tseng, Hung-Yu, et al. Regularizing Generative Adversarial Networks under Limited Data. CVPR 2021

---

> > > > ### Author Response · Authors · 2023-08-14
> > > > **Thanks for the Reply (2/2)**
> > > >
> > > > - **Concrete examples of how the  parameter $\omega$ is defined for various augmentations?**
> > > >
> > > > As our AugSelf-GAN relies entirely on data augmentation used in DiffAugment, we would like to use the three data augmentation employed by DiffAugment as an example to answer your question. DiffAugment uses color, translation, and cutout as data augmentations.
> > > >
> > > > For color, there are three parameters: $\lambda_\text{brightness}$, $\lambda_\text{saturation}$, and $\lambda_\text{contrast}$, which are randomly sampled from the range $[0,1]$. Therefore, we directly combine them as the parameter $\omega_\text{color}\in [0,1]^3$.
> > > >
> > > > Translation includes two parameters, $x_\text{translation}$ and $y_\text{translation}$, which represent how many pixels the image is translated on the x and y axis, respectively. In the DiffAugment code, they are randomly sampled from the integer range {$-X_\text{translation},\dots,X_\text{translation}$} and {$-Y_\text{translation},\dots,Y_\text{translation}$}, respectively. We transform them into the range of $[0,1]^2$ through a linear transformation ($(x_\text{translation} / X_\text{translation} + 1) / 2$, $(y_\text{translation} / Y_\text{translation} + 1) / 2$) as the parameter $\omega_\text{translation}$.
> > > >
> > > > Cutout includes two parameters, $x_\text{offset}$ and $x_\text{offset}$, which represent the center coordinate of the cutout.They are randomly sampled from the integer range {$\{0,\dots,X_\text{offset}\}$} and {$\{0,\dots,Y_\text{offset}\}$}, respectively. Similar to translation, we transform them into the range of $[0,1]^2$ through a linear transformation ($x_\text{offset} / X_\text{offset}$, $y_\text{offset} / Y_\text{offset}$) as the parameter $\omega_\text{cutout}$.
> > > >
> > > > For the specific implementation of the above scheme, please refer to the `DiffAugment_pytorch.py` file in the anonymous code repository provided in our paper.
> > > >
> > > > Thank you for your time and effort in reviewing our manuscript. If you have any further questions or suggestions, we would be happy to hear them.
> > > >
> > > > Best regards,
> > > >
> > > >  Authors of Submission 5698

---

> > > > > ### Comment · Reviewer_nM6o · 2023-08-20
> > > > >
> > > > > Thanks for the response. However, there are still two concerns that I have:
> > > > >
> > > > > 1) Is it optimal to combine different augmentations with the same value of $w$? If the value of $w = 0.5$ for both the translation and cutout, are they equivalent augmentations? If not, then why should the discriminator predict the same for both?
> > > > > I don't understand the reasoning behind the same $w$ parameter for all augmentations. Hence, I am unsure of the correctness of the above procedure.
> > > > >
> > > > >
> > > > > 2) As the AHM divergence is shown to be the major reason for performance improvement. Can the authors give some concrete evidence regarding experiments to verify the claim regarding AHM divergence?

---

> > > > > > ### Author Response · Authors · 2023-08-21
> > > > > > **Response by Authors**
> > > > > >
> > > > > > Dear Reviewer nM6o,
> > > > > >
> > > > > > Thank you for your feedback on our paper. We sincerely apologize for any confusion or misunderstanding caused by our insufficient description of our method. We appreciate the opportunity to clarify the details of our approach.
> > > > > >
> > > > > > **The self-supervised labels $\mathbf{\omega}$ in different augmentations**
> > > > > >
> > > > > > Regarding the use of color, translation, and cutout as self-supervised signals, we would like to clarify that we employ separate self-supervised discriminators for each task, and they are not shared. In other words, our method includes four losses for the discriminator component: one for the original discriminator $D$ and one each for the color, translation, and cutout self-supervised discriminators ($\hat{D}\_\text{color},\hat{D}\_\text{translation},\hat{D}\_\text{cutout}$). These four networks share a common backbone $\phi$ and differ only in the final layer ($\psi,\varphi\_\text{color},\varphi\_\text{translation},\varphi\_\text{cutout}$), respectively. Therefore, different augmentations do not share the same $\mathbf{\omega}$, as can be seen from lines 122-124, where the dimensions of $\mathbf{\omega}\_\text{color}\in\mathbb{R}^3$ and $\mathbf{\omega}\_\text{translation}\in\mathbb{R}^2$ are not the same.
> > > > > >
> > > > > > We will rewrite Eqs. 7 and 8 as the following to make the details of our method more clear.
> > > > > > $$
> > > > > > \min\_{D,\hat{D}\_\text{color},\hat{D}\_\text{translation},\hat{D}\_\text{cutout}} \mathcal{L}\_D^\text{da}+\lambda\_d\cdot\left(\mathcal{L}\_{\hat{D}\_\text{color}}^\text{color}+\mathcal{L}\_{\hat{D}\_\text{translation}}^\text{translation}+\mathcal{L}\_{\hat{D}\_\text{cutout}}^\text{cutout}\right)
> > > > > > $$
> > > > > > $$
> > > > > > \min\_{G} \mathcal{L}\_G^\text{da}+\lambda\_g\cdot\left(\mathcal{L}\_{G}^\text{color}+\mathcal{L}\_{G}^\text{translation}+\mathcal{L}\_{G}^\text{cutout}\right)
> > > > > > $$
> > > > > > where $\mathcal{L}^{\text{color}}\_{\hat{D}\_\text{color}}$ and $\mathcal{L}^{\text{color}}\_G$ are defined as follows:
> > > > > > $$
> > > > > > \mathcal{L}^{\text{color}}\_{\hat{D}\_\text{color}} = \mathbb{E}\_{\mathbf{x},\boldsymbol{\omega}}\left[\|\hat{D}\_\text{color}(T(\mathbf{x};\boldsymbol{\omega}),\mathbf{x})-\boldsymbol{\omega}\_\text{color}^+\|\_2^2\right] + \mathbb{E}\_{\mathbf{z},\boldsymbol{\omega}}\left[\|\hat{D}\_\text{color}(T(G(\mathbf{z});\boldsymbol{\omega}),G(\mathbf{z}))-\boldsymbol{\omega}\_\text{color}^-\|\_2^2\right]
> > > > > > $$
> > > > > > $$
> > > > > > \mathcal{L}^{\text{color}}\_G = \mathbb{E}\_{\mathbf{z},\boldsymbol{\omega}}\left[\|\hat{D}\_\text{color}(T(G(\mathbf{z});\boldsymbol{\omega}),G(\mathbf{z}))-\boldsymbol{\omega}\_\text{color}^+\|\_2^2\right] - \mathbb{E}\_{\mathbf{z},\boldsymbol{\omega}}\left[\|\hat{D}\_\text{color}(T(G(\mathbf{z});\boldsymbol{\omega}),G(\mathbf{z}))-\boldsymbol{\omega}\_\text{color}^-\|\_2^2\right]
> > > > > > $$
> > > > > > $\mathcal{L}^{\text{translation}}\_{\hat{D}\_\text{translation}}, \mathcal{L}^{\text{cutout}}\_{\hat{D}\_\text{cutout}}, \mathcal{L}^{\text{translation}}\_G, \mathcal{L}^{\text{cutout}}\_G$ follow the similar definition. And $\mathbf{\omega}$ represents all augmentations, i.e., data $\mathbf{x}$ is sequentially transformed using color, translation, and cutout augmentations.
> > > > > >
> > > > > >
> > > > > > **Concret experimental evidence on AHM divergence**
> > > > > >
> > > > > > We are very grateful for your constructive questions. However, it is difficult to directly verify the feedback robustness of AHM through experiments, as we cannot truly construct and obtain the results of the f function. Nevertheless, based on current experiments, we can assist in verifying the advantages of our method from the loss curve of the discriminator in data-limited scenarios. Specifically, the lower the loss in the discriminator, the more obvious the overfitting phenomenon, and the **higher** it indicates that **overfitting can be alleviated**. Unfortunately, we are no longer able to upload the loss curve through a PDF. According to the existing experimental records, as shown in the table below, the average value of the training loss on the DiffAugment discriminator is significantly smaller than that on our AugSelf-GAN, indicating that AugSelf-GAN can effectively further alleviate the overfitting of DiffAugment.
> > > > > >
> > > > > > | D_loss | CIFAR-10 | CIFAR-10 20% | CIFAR-10 10% | CIFAR-100 | CIFAR-100 20% | CIFAR-100 10% |
> > > > > > | --- | --- | --- | --- | --- | --- | --- |
> > > > > > | DiffAugment | 1.0902 | 0.7157 | 0.6964 | 1.0474 | 0.6404 | 0.8974 |
> > > > > > | AugSelf-GAN | **1.4564** | **1.2364** | **1.1331** | **1.3818** | **1.1073** | **1.1719** |
> > > > > >
> > > > > > Thank you again for your coment. We sincerely hope that our response can address your question.

---

### Official Review · Reviewer_b4Yw · 2023-07-08

**Soundness:** 3 good
**Presentation:** 4 excellent
**Contribution:** 3 good
**Rating:** 5
**Confidence:** 5

**Summary:**

This work proposed an augmentation aware training framework to improve the synthesis of GANs under the low data regimes. The discriminator is also required to regress the augmentation parameters, preventing undesired invariance introduced by augmentations. Theoretical analysis also shows the connection between the given objective and the arithmetic-harmonic mean divergence. Experimental results alos demonstrate the effectiveness of the proposed approach under various benchmarks.

**Strengths:**

- This paper is well-organized and easy to follow.
- Making discriminator aware of the appiled augmentation is inspiring since some undesired invariance is always introduced more or less. - - Experiments are quite solid, including results on multiple benchmarks and insightful analysis (Sec. 6.2).
- Section 5 also presents the theoretical analysis, showing the connection with arithmetic harmonic mean divergence, which may gives more deep understanding for audience.

**Weaknesses:**

- There might lack the comparisons against prior SOTA. For instance, InsGEN and Vision-aided GAN achieves appealing performances for FFHQ and while Table.2 did not presents these. Obviously, the consistent gains could be obtained compared to the baseline StyleGANs. However, SOTA performances could make this work stronger.
- Experiments on AFHQ are missing. I believe AFHQ is also a good benchmark to evaluate the approaches for limited data synthesis.
- Performances on the sufficient data for FFHQ and LSUN-Cat would further give a whole picture of how well the proposed method performs.
- Could this augmentation aware scheme be combined with more complex augmentations like ADA?

**Questions:**

Please see Weaknesses

**Limitations:**

Please see Weaknesses

---

> ### Author Rebuttal · Authors · 2023-08-09
>
> We would like to thank Reviewer b4Yw for the valuable feedback. The comments and suggestions have greatly helped us in improving the quality of our work and ensuring its scientific rigor.
>
> ---
>
> **There might lack the comparisons against prior SOTA (InsGEN and Vision-aided GAN).**
>
> Thank you very much for your suggestion. We would like to further clarify the motivation behind our work to demonstrate the rationale of our experimental design in the initial version of our paper.
>
> First, we do not consider Vision-aided GAN as a baseline as it introduces additional pre-trained models and data, and is a fine-tuning work based on our baseline, DiffAugment. Therefore, this comparison would be unfair. InsGen is an improvement on ADA, which utilizes richer data augmentation techniques than DiffAugment with a total of 18 transformations. Therefore, InsGen may achieve additional gains due to more data augmentation. Of course, our AugSelf-GAN can also be applied to ADA's 18 transformations, which will include discrete 90-degree rotation data augmentation. However, this would partially include SSGAN-LA, which is not our intended result.
>
> DiffAugment is our direct baseline, and our goal is to surpass it. We have verified this through a large number of fair experiments. After this, we once again thank you for your suggestion and acknowledge that SOTA's performance will make this work even stronger. Therefore, we will consider this as our future work, exploring extending AugSelf-GAN to more powerful data augmentations and compare it with SOTA methods.
>
> ---
>
> **Experiments on AFHQ are missing.**
>
> We appreciate your thoughtful suggestion to evaluate our proposed method on AFHQ, which is a suitable dataset for limited data scenario. We are currently conducting experiments on AFHQ, but due to limited training resources, it is taking us some time to obtain reliable results. However, we assure you that we will include the results in the updated version of the paper.
>
> ---
>
> **Performances on the sufficient data for FFHQ and LSUN-Cat.**
>
> Thank you for your suggestion regarding the evaluation on the full FFHQ and LSUN Cat datasets. However, we apologize that conducting such experiments and obtaining results during the short rebuttal phase would be difficult for us. We will continue to conduct experiments on full FFHQ and LSUN Cat data and report the results. We chose the current data setting in experiments because our research focuses on training GANs with **limited data.** Therefore, we follow exactly the same data setting as our baseline DiffAugment.
>
> ---
>
> **Could this augmentation aware scheme be combined with more complex augmentations like ADA?**
>
> Thank you for your constructive question, which enriched our experiments and improved our work. We believe our method can indeed be combined with ADA. One major difference between ADA and DiffAugment is that ADA uses adaptive data augmentation probabilities. In fact, ADA incorporates the same augmentations as DiffAugment, such as brightness, contrast, saturation, translation, and cutout. We still use these as self-supervised signals for our AugSelf framework to conduct new experiments based on ADA on unconditional CIFAR10, and the results are as follows, showing that our method outperforms ADA.
>
> | FID | uncond CIFAR-10 |
> | --- | --- |
> | ADA | 3.11 |
> | AugSelf | **3.02** |
>
> We will continue to conduct more experiments, including more data augmentation as self supervision, to verify the applicability of our method in stronger data augmentation methods.

---

### Official Review · Reviewer_CpCq · 2023-07-08

**Soundness:** 2 fair
**Presentation:** 3 good
**Contribution:** 2 fair
**Rating:** 5
**Confidence:** 2

**Summary:**

This paper addresses the challenge of training GANs with limited data, where the discriminator is prone to overfitting. Previous approaches, such as differentiable augmentation, have improved data efficiency but introduce undesired invariance to augmentation for the discriminator. To overcome this limitation while retaining the benefits of data augmentation, the authors propose an augmentation-aware self-supervised discriminator. This discriminator predicts the augmentation parameter of the augmented data, distinguishing between the prediction targets of real and generated data during training. The generator is encouraged to learn from the self-supervised discriminator by generating augmentation-predictable real data. This formulation connects the learning objective of the generator and the arithmetic-harmonic mean divergence.

**Strengths:**

This paper proposes an augmentation-aware self-supervised discriminator. The authors include both experimental results and theoretical analysis to show the effectiveness of the proposed method. According to the results, AugSelf can improve the performance than SOTA methods.

**Weaknesses:**

The major concern is the limited novelty. For example, SSGAN-LA works on augmenting the GAN labels (real or fake) via self-supervision of data transformation. The only difference is the form of self-supervised signals. Though the authors try to highlight the differences between these two works, these two methods still share many similarities. In the experiments, the paper also doesn't include a direct comparison with the closely related work SSGAN-LA.

Furthermore, the method relies on the used augmentation method. How to select the appropriate augmentation is a problem. The paper doesn't include guidelines of how to select these augmentations.

**Questions:**

1. Is there a way to determine which augmentations are beneficial to the training?

2. What are the major differences between this work and  SSGAN-LA ?

3. SSGAN-LA can addresses catastrophic forgetting problem. Is AugSelf also able to tackle this challenge?

**Limitations:**

The limitations are elaborated in the prior sections. In summary, the limited novelty is the major concern.

---

> ### Author Rebuttal · Authors · 2023-08-09
>
> We would like to thank Reviewer CqCq for providing valuable feedback. The comments and suggestions have greatly helped us in improving the quality of our work and ensuring its scientific rigor.
>
> ---
>
> **Limited novelty compared to SSGAN-LA and no direct comparison with SSGAN-LA.**
>
> AugSelf-GAN and SSGAN-LA have significant differences in the problems they aim to solve, their specific methods, and theoretical analysis. Firstly, in terms of the problem they aim to solve, SSGAN-LA focuses on improving the training stability of unsupervised GANs and correcting the theoretical learning objective deviation of existing self-supervised GANs. On the other hand, our AugSelf-GAN aims to enhance the performance of GANs in data-limited scenarios, specifically the efficiency of utilizing training data.
>
> In terms of methods, SSGAN-LA employs a discrete adversarial self-supervised task, while AugSelf-GAN utilizes a continuous adversarial self-supervised task. It should be noted that the transition from discrete to continuous self-supervised adversarial tasks is not easy. Specifically, the adversarial idea requires us to distinguish self-supervised labels of true and false data. In the discrete case, SSGAN-LA uses class augmentation to distinguish the categories of true and false data, while in the continuous case, there is no similar concept. Therefore, we must design a brand-new adversarial task that conforms to the continuous label. We innovatively choose the origin-symmetric scheme ($\omega^+=-\omega^-$) for distinguishing continuous labels of true and false data and provide corresponding theoretical explanations.
>
> This theoretical explanation reveals the correlation between the proposed method and a robust divergence, explaining the training stability of the method under data constraints. This advantage is not present in SSGAN-LA, whose theoretically optimized divergence is the reversed KL divergence that is inferior to the AHM divergence in this particular scenario.
>
> Additionally, in terms of the specific implementation of the discriminator, SSGAN-LA does not accept original data as input and thus requires the data augmentation to be reversible, while AugSelf-GAN accepts both original and augmented data as input, making it unrestricted in data augmentation. We believe that these significant differences demonstrate the novelty of this work compared to SSGAN-LA.
>
> The table below compares AugSelf-GAN and SSGAN-LA on the CIFAR-10 and CIFAR-100. According to the IS and FID scores, AugSelf-GAN outperforms SSGAN-LA in most settings, especially when the data is small. We appreciate your constructive suggestions, and will add this comparative experiment in the updated version of our paper.
>
> | IS, FID | CIFAR-10 | CIFAR-10-0.2 | CIFAR-10-0.1 | CIFAR-100 | CIFAR-100-0.2 | CIFAR-100-0.1 |
> | --- | --- | --- | --- | --- | --- | --- |
> | SSGAN-LA | 9.30, 8.05 | 8.84, 11.16 | 8.30, 15.08 | 11.13, **9.77** | **10.52**, 15.91 | 9.81, 23.17 |
> | AugSelf-GAN | **9.43**, 7.68 | 8.98, 10.97 | 8.76, 15.68 | **11.19**, 9.88 | 10.25, 16.11 | 9.78, 21.30 |
> | AugSelf-GAN+ | 9.27, **7.54** | **9.08**, **9.95** | **8.79**, **12.76** | 11.12, 10.09 | 10.14, **15.33** | **9.93**, **18.64** |
>
> ---
>
> **Is there a way to determine which augmentations are beneficial to the training?**
>
> According to Table 7 in the appendix, we found that any type of data augmentation (color, translation, cutout), as well as their combinations, are beneficial for GAN training compared to the baseline method DiffAugment. We believe that this is one of the advantages of AugSelf-GAN, which is effective and robust in selecting various types of data augmentation. For simplicity, we default to using all data augmentations of baseline methods (e.g. DiffAugment) as self-supervised signals. As for the strength of data augmentation, experiments in Table 5 show that fewer training data require stronger data augmentation, and vice versa. We appreciate your valuable feedback and will clarify this point in the updated version of our paper.
>
> ---
>
> **Is AugSelf also able to tackle catastrophic forgetting problem?**
>
> Thank you for your constructive question. We have conducted additional experiments and have provided the results in Figure 11 (d) and (h) of the new PDF file. AugSelf-BigGAN(+) achieved significantly better accuracy on the representation learning task in the discriminator compared to BigGAN and BigGAN+DiffAugment. The accuracy curve of AugSelf-BigGAN(+) tended to be stable after a rapid increase, unlike BigGAN which showed a significant decrease with increased training iterations. This indicates that the AugSelf method effectively addresses the catastrophic forgetting problem in the discriminator.

---

> > ### Comment · Reviewer_CpCq · 2023-08-21
> >
> > I thank the authors for providing the response and adding additional experiments. I have checked the author response and the reviews from other reviewers. I would maintain my ratings to accept the paper.

---

### Official Review · Reviewer_xecU · 2023-07-26

**Soundness:** 2 fair
**Presentation:** 3 good
**Contribution:** 2 fair
**Rating:** 5
**Confidence:** 3

**Summary:**

This paper deals with the challenge of data-efficient GAN training. It is built upon DiffAugment and attempts to mitigate its issue of undesired augmentation invariance. The proposed method introduces an additional linear layer that predicts the augmentation parameter from the features extracted by the backbone of the discriminator.

**Strengths:**

- The paper’s target task is clearly stated and holds significance.

- The proposed method is simple and exhibits consistent effectiveness across various scenarios.

**Weaknesses:**

- The motivation of the proposed method is unclear and lacks of evidence. The authors discuss the issue of DiffAugment in lines 36-42, citing [23] to argue the importance of augmentation sensitivity. However, it is questionable whether the motivations discussed in [23] are directly applicable to this work since the former aims at enhancing the transferability of the pretrained features. It is comprehensible that augmentation invariance can be harmful for the transferability of representation learning, and Figure 4 and 5 in [23] demonstrate intuitively that augmentation awareness helps the model learn position- and color-sensitive representations. However, in the case of image generation, it is hard for me to understand why the discriminator should be transformation sensitive.


- One of the major benchmark dataset, ImageNet, is not considered in this work, which is a common choice in GAN literature [1, 2, 3]. Given that the proposed method is mainly an improvement of DiffAugment, it would be vital to show the comparison on the primary dataset used in the original paper.

- The paper could benefit from a  qualitative comparison with the baselines by making it easier to intuitively grasp the effectiveness of the proposed method.


[1] Brock et al. Large scale GAN training for high fidelity natural image synthesis. 2019.

[2] Lucic et al. High-Fidelity Image Generation With Fewer Labels. 2019.

[3] Zhao et al. Differentiable augmentation for data-efficient gan training. 2020.

**Questions:**

Can you provide more explanation about how the theoretical analysis that “reveals a connection between the optimization objective of the generator and the arithmetic harmonic mean divergence” is relevant to the proposed method?

**Limitations:**

The authors have discussed the limitations of their work. Please see the Weaknesses stated above to see other limitations I find.

---

> ### Author Rebuttal · Authors · 2023-08-09
>
> We would like to thank Reviewer xecU for the valuable feedback. The comments and suggestions have greatly helped us in improving the quality of our work and ensuring its scientific rigor.
>
> ---
>
> **The motivation of the proposed method is unclear and lacks of evidence.**
>
> We apologize for any confusion caused by our explanation of the motivation behind our method. Please allow us to provide additional clarification.
>
> The sensitivity of the discriminator to different levels of data augmentation enables it to learn rich data representation such as color and object position information, thus enhancing the discriminator's representation learning ability. This has been supported by Table 2, where AugSelf-GAN exhibits stronger discriminator representation learning ability than DiffAugment. As previous works have shown [1,2,3,4], the stronger representation learning ability of the discriminator leads to better feedback to the generator, resulting in better fidelity and diversity of generated samples, which is consistent with our experimental results. In summary, we enhance the representation learning ability of the discriminator through perceptual augmentation-based self-supervised tasks, resulting in higher quality feedback to the generator and thus improved performance.
>
> We appreciate your feedback and will revise our manuscript to make our motivation and evidence clearer.
>
> ---
>
> **One of the major benchmark dataset, ImageNet, is not considered in this work.**
>
> We appreciate your suggestion to include ImageNet experiments in our research. While we understand that ImageNet is a common choice in GAN literature, we did not conduct ImageNet experiments initially due to resource limitations. Our code is mainly based on the open-source data-efficient-gans repository that builds on top of DiffAugment. The ImageNet on BigGAN experiments require TPUs, which we are not able to access.
>
> Nonetheless, we conducted high-resolution (256x256) experiments using StyleGAN2 and are currently in the process of conducting ImageNet experiments. Regrettably, because training BigGAN on ImageNet requires eight GPUs running for approximately a month to obtain good results, we are unable to report the results within the short rebuttal period. We are, however, working to reproduce the ImageNet experiments on GPUs and will add the results to an updated version of the paper once they are available.
>
> ---
>
> **The paper could benefit from a qualitative comparison with the baselines.**
>
> We have uploaded a PDF file to show the qualitative comparison of AugSelf-GAN with the baselines on CIFAR-10, Low-shot, FFHQ and LSUN-Cat (Figure 12, 13, 14), and will include it in the updated version of our paper. Please review the new PDF file, hopefully it will address your concerns.
>
> ---
>
> **How the theoretical analysis is relevant to the proposed method?**
>
> Our theoretical analysis starts from the objective function of the proposed method and is based on an assumption (that the self-supervised signal $\omega^+=-\omega^-$ is constant). As GANs solve a bi-level optimization problem, we first analyze the mathematical form of the optimal self-supervised discriminator and then substitute it into the objective function of the generator, resulting in the generator optimizing the arithmetic harmonic mean divergence. Although this assumption is different from our actual approach (where $\omega^+=-\omega^-$ is not constant), it is not uncommon in the field, as seen in non-saturating GAN [5] and LeCam-GAN [6]. Furthermore, we claim to reveal a connection rather than equivalence between the optimization objective of the generator and the arithmetic harmonic mean divergence. In our experiments, we demonstrate the effectiveness of dynamic self-supervised signals compared to fixed/constant ones.
>
> ---
>
> [1] Chen, et al. Self-Supervised GANs via Auxiliary Rotation Loss. CVPR 2019
>
> [2] Jeong, et al. Training GANs with Stronger Augmentations via Contrastive Discriminator. ICLR 2021
>
> [3] Sauer, et al. Projected GANs Converge Faster. NeurIPS 2021
>
> [4] Kumari, et al. Ensembling off-the-shelf Models for GAN Training. CVPR 2022
>
> [5] Goodfellow, et al. Generative Adversarial Nets. NIPS 2014
>
> [6] Tseng, et al. Regularizing Generative Adversarial Networks under Limited Data. CVPR 2021

---

> > ### Comment · Reviewer_xecU · 2023-08-14
> > **Thanks for your response**
> >
> > Thank you for your effort to respond. I think my concerns are well addressed except that the lack of ImageNet experiments could be a major flaw. I have raised my score to borderline accept.

---

### Author Rebuttal · Authors · 2023-08-09

We would like to sincerely thank all the reviewers for their valuable time and constructive feedback that have helped improve the quality of our research.

We have carefully considered all the concerns and questions raised by the individual reviewers and provided detailed responses in our rebuttal. We hope that the reviewers and the AC will take our responses into account while evaluating our manuscript.

To further demonstrate the effectiveness of our proposed method, we have uploaded a PDF file containing several figures. Figure 11 shows the convergence comparison between our method and the baselines on FID plots (a,b,c,e,f,g) and Acc plots (d,h) to validate the training convergence and ability of our method to overcome catastrophic forgetting, respectively. Figures 12, 13, and 14 exhibit qualitative comparisons between our proposed method AugSelf and DiffAugment on CIFAR-10, CIFAR-100, Low-shot image generation, FFHQ, and LSUN-Cat with limited data.

Finally, we would like to express our gratitude to everyone for their efforts and contributions to this conference. We hope that our paper can be fully discussed in the next stage.

---

### Decision · Program_Chairs · 2023-09-21

**Decision:**

Accept (poster)

**Comment:**

5x BA. This paper proposes an augmentation-aware self-supervised discriminator for low-shot GAN training, so as to mitigate the discriminator overfitting and improve the generation quality. The reviewers all lean to accept the paper due to its (1) clear presentation, (2) effective empirical results, and (3) comprehensive theoretical analysis. The rebuttal has addressed their concerns.